# GCAM v5.1: Representing the linkages between energy, water, land, climate, and economic systems

Katherine Calvin,[1] Pralit Patel,[1] Leon Clarke,[1] Ghassem Asrar,[1] Ben Bond-Lamberty,[1] Alan Di Vittorio,[2] Jae Edmonds,[1] Corinne Hartin,[1] Mohamad Hejazi,[1] Gokul Iyer,[1] Page Kyle,[1] Sonny Kim,[1] Robert Link,[1] Haewon McJeon,[1] Steven J Smith,[1] Stephanie Waldhoff,[1] Marshall Wise[1]

[1]Pacific Northwest National Laboratory's Joint Global Change Research Institute, College Park, MD, USA
[2]Lawrence Berkeley National Laboratory, Berkeley, CA, USA

*Correspondence to*: Katherine Calvin (katherine.calvin@pnnl.gov)

**Abstract.** This paper describes GCAM v5.1, an open source model that represents the linkages between energy, water, land, climate, and economic systems. GCAM is a market equilibrium model, is global in scope, and operates from 1990 to 2100 in five-year time steps. It can be used to examine, for example, how changes in population, income, or technology cost might alter crop production, energy demand, or water withdrawals, or how changes in one region's demand for energy affect energy, water, and land in other regions. This paper describes the model, including its assumptions, inputs, and outputs. We then use eleven scenarios, varying socioeconomic and climate policy assumptions, to illustrate the results from the model. The resulting scenarios illustrate a wide range of potential future energy, water, and land uses. We compare the results from GCAM v5.1 to historical data and to future scenario simulations from earlier versions of GCAM and from other models. Finally, we provide information on how to obtain the model.

## 1 Introduction

Researchers and decision makers are increasingly interested in understanding the many ways in which human and Earth systems interact with one another, at scales from local (e.g., a city) to regional to global (Palmer and Smith, 2014). For example, how might new emerging technologies such as photovoltaic cells or new batteries influence the way that energy is consumed and used, and what might this mean for greenhouse gas emissions, air pollution, international markets for fossil fuels, and access to energy? How might changes in population, income, or technology cost alter crop production, energy demand, or water withdrawals? How do changes in one region's demand for energy affect energy, water, and land in other regions?

A number of modeling tools and frameworks have been established to explore questions such as these, representing the multiple interactions among human and Earth systems (Calvin and Bond-Lamberty, 2018; Weyant, 2017). This paper introduces the most recent version of one such model, GCAM v5.1. GCAM represents the behavior of, and complex interactions between five major systems – energy, water, land, climate, and the economy – at global and regional scales. GCAM simulates changes in these systems for decades into the future. GCAM has its roots in the Edmonds-Reilly model (Edmonds and Reilly, 1983b,

1983a, 1983c) developed in the late 1970s and early 1980s and has been continuously updated since then (Kim et al., 2006; Wise et al., 2009, 2014a). Over time, as scientific questions have become more complex, GCAM has also evolved in complexity, transitioning from a focus solely on energy and $CO_2$ emissions to an examination of questions at the intersection of energy, water, land, socioeconomics, and climate.

The model represents all of these systems in a single, integrated computational platform rather than linking among models operating in different platforms, although some components (e.g., the climate system) can be run individually as well. This allows insights that are not possible in single sector or single system models. Models such as GCAM are designed to answer what if questions about the future; that is, they help us understand how the future will evolve under a particular set of conditions

and how the system will change under the influence of external factors. For example, users can examine the influence of changes in socioeconomics or policy on energy, water, and land in GCAM (as shown in Section 4). GCAM can also be used to explore the implications of changes in one region on other regions (e.g., Wise et al. (2014b)).

GCAM is computationally inexpensive[1] enabling the exploration of multiple scenarios (Calvin et al., 2014, 2017; Graham et

al., 2018) and large ensembles (Lamontagne et al., 2018) to develop robust insights given the significant uncertainty in future conditions. Individual component modules in GCAM are designed to capture key characteristics of the underlying systems; however, because its focus is on the interactions among systems, it does not include the level of detail found in sector- or process-specific models.

There are a number of models in the community with similar overall scope to GCAM, although each has a unique structure and focus. The IMAGE (Stehfest et al., 2014) model also contains a dynamic-recursive energy system module, which is soft-linked with a detailed land-use module, representing a different tradeoff between integration and physical detail. There are a number of models with some form of inter-temporal optimization such as DNE21 (Akimoto et al., 2010), REMIND (Kriegler et al., 2017), MESSAGE-GLOBIOM (Fricko et al., 2017), and WITCH (Bosetti et al., 2007). Economic structures also vary,

from partial-equilibrium for MESSAGE-GLOBIOM and IMAGE to computable general equilibrium for AIM/CGE (Fujimori et al., 2017) and ENV-Linkages (Chateau et al., 2014). Several recent papers provide comparisons of GCAM to these models and many others, including discussions of model structure, input data, and results (Bauer et al., 2018; Popp et al., 2017; Rao et al., 2017).

In the remainder of this paper, we provide an introduction to and an overview of GCAM v5.1, released on July 9, 2018. Section 2 describes the model, including each of its component parts. Section 3 describes some simulations to illustrate the capabilities

---

[1] A single 100-year simulation using GCAM runs in 10-15 minutes on a laptop. More complex options, e.g., limiting radiative forcing to a particular level, requires numerous sequential 100-year simulations increasing the run time.

of the model. Model results are highlighted in Section 4. Section 5 is focused on discussion and conclusions. Finally, Section 6 provides information on how to obtain the model. This paper provides a general overview of the model. More detailed documentation is available on-line at http://jgcri.github.io/gcam-doc.

## 2 Model Description

### 2.1 Overview of GCAM

GCAM represents five different interacting and interconnected systems: energy, water, land, socioeconomics, and climate (Figure 1). These systems are represented at a variety of spatial scales (Figure 2). For example, economic and energy systems are represented at 32 geopolitical regions, which is sufficient to gain many insights about broad international socioeconomic and energy dynamics. The land and water system, however, is subdivided into water basins (resulting in 384 land-water regions), because of the need to link water and agriculture in order to effectively represent the interactions between these systems (e.g., the implications of future droughts, or the hydrological implications agriculture production).

### 2.2 The GCAM Core

The GCAM core is the model component where the economic decisions and dynamic interactions between the various systems are represented (Figure 3). It is written in C++, uses XML input files, and generates a hierarchical output database.

The operating principle for GCAM is that of market equilibrium. Representative agents in GCAM use information on prices, costs, and other relevant factors to make decisions about the allocation of resources. These representative agents exist throughout the model, for example for regional electricity sectors, regional refining sectors, regional energy demand sectors (e.g., a representative residential building), and agents who allocate land among competing uses within any given region. Markets are the means by which these representative agents interact with one another. Agents pass supply and demand for goods and services into the markets. Markets exist for physical flows such as electricity or agricultural commodities, but they also can exist for other types of goods and services, such as tradable emissions permits. GCAM solves for a set of market prices such that supplies and demands are equal for all markets in the model. The GCAM solution process involves iterating on market prices until this equilibrium is reached within a user-specified tolerance level.

As an example of this process, in any single model period, GCAM derives a demand for natural gas starting with all of the uses to which natural gas might be put, such as passenger and freight transport, power generation, hydrogen production, heating, cooling and cooking, fertilizer production, and other industrial energy uses. Those demands depend on the external assumptions about, for example, electricity generating technology efficiencies, but also on the price of all of the commodities in the model. GCAM computes the supplies of all of the goods and services in the model. For example, it calculates the amount of natural gas that suppliers would like to supply given their available technology for extracting resources, and the market

price. The model sums all of the supplies and demands for commodities and adjusts prices, so that in every market during that period supplies of everything from rice to solar power match demands.

GCAM is a dynamic recursive model, i.e., decision-makers base their decisions only on currently available information rather than optimizing over the full future, as is the case in intertemporal optimization models. For long-lived capital stocks, decision-makers in GCAM factor in potential future costs and revenues but do this assuming today's market prices. After it solves each period, the model uses the resulting state of the world, including the consequences of decisions made in that period (e.g., resource depletion, capital stock retirements and installations, changes to the landscape, emissions into the atmosphere), as a starting point for the next time step.

Decision-making throughout GCAM uses a logit formulation (Clarke and Edmonds, 1993; McFadden, 1973). In such a formulation, options are ordered based on preference, with either cost (as in the energy system) or profit (as in the land system) determining this ordering. However, the single best choice does not capture the entire market. A variety of factors not captured in the model, such as individual preferences, local variations in cost/profit, and simple happenstance, cause some of the market to go to alternatives that, based on their cost or profit alone, are theoretically inferior choices. The exact share a given option receives in GCAM depends on the logit exponent and the share weight:

$$s_i = \frac{\alpha_i c_i^\gamma}{\sum_{j=1}^N \alpha_j c_j^\gamma}$$

where $s_i$, $c_i$, $\alpha_i$ are the share, cost, and share weight of technology i, respectively, and $\gamma$ is the logit exponent.[2] Logit exponents are exogenously-specified and dictate the degree to which cost or profit determines share; exponents that are larger in absolute magnitude result in more winner-take-all behavior. Share weights are mostly calculated in the historical period to ensure that GCAM replicates historical data; however, these are on occasion over-written in future periods, for example, to represent a scenario where a now relatively new technology becomes widely available. The general philosophy in GCAM is to maintain share weights at their calibrated values (which ensure the model replicates history) unless:

(1) We consider that the past is not a good analog for the future, as with emerging technologies (e.g., solar, wind) where information barriers or lack of infrastructure may prevent their adoption today, but those factors will likely be ameliorated with time, or

(2) The specific scenario being produced necessitates changes to the share weights (e.g., solar and wind in the SSP1).

## 2.3 The GCAM Data System

The GCAM data system produces inputs for the dynamical core. The data system is written as an easily-installable R package, uses CSV input files, and generates XML files used as inputs to the GCAM core. These files contain both historical information

---

[2] For some sectors, GCAM uses a slight variation on this formula (see http://jgcri.github.io/gcam-doc/choice.html).

used to initialize GCAM, as well as parameters that govern changes in the future. GCAM is an input-driven model (Kim et al., 2006) where the specific model regions, sectors, and technologies are dynamically created as their data specifications are parsed. As a result, many user changes are implemented via changes to the data system and do not require changes in the dynamical core.

For historical information, the GCAM data system starts with country-level inventory data on energy production and consumption, agricultural production and consumption, land use and land cover, water demand, and emissions of 24 species. These data are aggregated to GCAM regions, commodities, and sectors. Adjustments are made to the data as needed to fill in missing information and to ensure that supplies and demands balance during the historical period (1990-2010). For the future parameters, the GCAM data system uses information about population and labor productivity, information about technology cost and performance, information about resource bases, and information on non-$CO_2$ mitigation potential. These data are also aggregated and mapped to GCAM regions, technologies and sectors.

Additional information on the data used in each module is described in the sections below. Additional information on the data system's code and processing steps is given in Bond-Lamberty et al. (2018).

## 2.4. Major Changes from Previous Versions

Over time, GCAM has evolved to incorporate new features and more detail, such as more detailed land use (starting with GCAM v3), increased regional resolution (starting with GCAM v4), and incorporating water demand (starting with v5). The most recent updates (relative to GCAM v4) include:

- Incorporating water demands,
- Changing the land regions to be based on water basins, instead of agro-ecological zones,
- Including multiple agricultural management practices, which enables intensification,
- Including five alternative socioeconomic pathways,
- Updating to a newer version of the climate model, and
- Including a new data processing system.

## 2.5 Socioeconomics

The scale of human systems in GCAM is set by two variables, population and the Gross Domestic Product (GDP). Population is an externally prescribed input to the model. GCAM requires values for population for each of the 32 geopolitical regions in each simulation period, both historical and future. GDP in each region and each period is a function of the previous period's GDP, the size of the labor force, and the labor productivity growth rate for that period. The size of the labor force is determined by the population size and the exogenously-specified labor force participation rate. The labor productivity growth rate is an

externally prescribed value, which measures inflation-adjusted growth in the value of goods and services produced, the method originally used by Edmonds and Reilly (1983a). The initial, historical year GDP value is a model input. At present socioeconomic variables, population and GDP, are independent of other GCAM components. That is, while population and GDP are determinants of activity levels in energy, water, land, and climate modules, activities in those sectors do not influence either population or GDP in GCAM v5.1.

Population data is from the Shared Socioeconomic Pathway 2 (SSP2), "Middle-of-the-Road," scenario, as developed by KC and Lutz (2017). Initial year GDP, labor productivity growth rates, and labor force participation rates are derived to match external GDP data and forecasts from three sources: 1) the USDA (2015) for 1990, 2005, and 2010; 2) the International Monetary Fund (2014) for 2011 to 2020; and 3) Delink et al. (2017), using the SSP2, for 2021 through 2100.

The two primary outputs are population and GDP by region and time period. GDP is provided in constant United States dollars for all regions. For non-U.S. regions, GDP is available at either market exchange rates (MER) or purchasing power parity (PPP).

## 2.6 Energy

The energy system of GCAM includes a comprehensive representation of energy production, transformation, distribution, and use, in each of 32 geopolitical regions. It starts with the resource bases of nine primary fuels in each region, the outputs of which pass through a series of energy handling, transformation, and distribution processes, finishing with the consumption of primary and final energy commodities by end-use sectors. The fundamental drivers of the energy system in each region are the population and GDP, which set the scale of the demands in the end-use sectors. Along any energy supply chain, the outputs of each modeled process are the inputs to the next.

For most primary fuels, resource production is modeled with exogenous supply curves, which prescribe the availability of energy production as a function of the energy price. Resources may be renewable (e.g., wind, solar), or depletable (e.g., fossil fuels and uranium). Renewable resource supply curves are indicated in EJ per year, whereas depletable resources are indicated as cumulative resource quantities (in EJ), which are drawn down in each time period as each resource is consumed. Resource costs, including depletion-related increases in fossil resource prices, may be counter-acted by exogenous technical change, which lowers extraction costs.

Aside from primary resource production, each sector, or process, in the energy system is represented with explicit technologies that consume inputs and produce outputs that then serve as inputs to other sectors. For example, the energy transformation sectors include a variety of technologies representing different electricity generation facilities (including different fuel sources and different technologies), different refineries (e.g., petroleum, bioliquids, coal-to-liquids, gas-to-liquids), different gas

processing facilities, and different hydrogen production facilities. Each technology is specified with a different set of inputs, costs, and performance characteristics. End-use demands form the end-point of the modeled energy supply chains. Structurally, each sector consists of at least one subsector, each of which has at least one technology. At both the subsector and technology levels, multiple options may compete for share on the basis of the relative costs, as well as preferences which are calibrated from historical choices. The market share of each technology within a subsector, or for each subsector within a sector, is endogenous based on the logit choice formulation.

Subsector costs are computed as the output-weighted average of technology costs, and sector costs are computed as the weighted average of subsector costs. As such, the fundamental determinant of the cost of each modeled sector (i.e., commodity, or market good) is the weighted average cost of its production technologies. These are computed as the sum of three explicitly represented cost components, each of which is indicated in dollars per unit of output: energy-input costs (i.e., the sum of the costs of all modeled inputs to the technology), exogenous "non-energy-input" costs (e.g., amortized capital costs and operations and maintenance costs), and ancillary costs such as emissions penalties. The costs of each energy-input are equal to the price of the relevant commodity multiplied by its exogenous input-output coefficient. Ancillary costs are specific to the policy type; for policies with an emissions price, the additional cost is equal to the emissions price multiplied by the amount of emissions of the specified species released per unit of output. Emissions prices can be exogenously specified or generated by the model if a constraint or target is imposed; these prices can vary across time, region, and gas.

Technologies in the energy system may produce emissions of a variety of species. The $CO_2$ emissions are computed as the sum of each energy-input times its exogenous carbon content, minus the fuel carbon content of the output fuel (if non-zero). For technologies with carbon capture and storage (CCS), or that are otherwise assumed to sequester carbon for a long time (e.g., industrial feedstocks), the amount sequestered is also deducted from the reported emissions.

Non-$CO_2$ emissions from any modeled technology may include greenhouse gases (e.g., $CH_4$, HFCs) and air pollutants (e.g., CO, $NO_x$, black carbon). The emissions of each species in each region, technology, and time period is computed as the technology's output multiplied by an emissions factor, which is generally derived from historical data in the model calibration years. In future years the emission factor may evolve as control rates change in response to growth in per-capita GDP and/or carbon pricing. This allows, for instance, reduction in the emissions factors of pollutants as countries become more wealthy (e.g., (Smith, 2005)), and reduction in emissions factors of greenhouse gases in response to climate policies (e.g., (EPA, 2013)). This approach is designed to capture general trends in emissions factors, but does not explicitly represent individual technologies or policies that may be adopted.

The primary data source for all energy flow volumes in the historical years is the IEA Energy Balances (IEA, 2012), which is used for calibration of energy production, transformation, energy losses in distribution, and consumption. Global production

and consumption volumes of coal, gas, and oil are scaled to remove any statistical differences and net stock changes, and electricity demand volumes are similarly scaled within each region so as to remove any net trade and statistical differences.

Primary resource supply curves for coal, gas, and oil are from Rogner (1997). Wind and distributed solar photovoltaic (PV) supply curves are from Zhou et al. (2012) and Denholm (2008), respectively. Supply curves for municipal waste-derived biomass energy are from Gregg and Smith (2010); other sources of biomass energy are supplied by the land component and discussed later. Hydropower is modeled as an exogenous output in all future years; the quantities are based generally on economic and technical potentials estimated by the International Hydropower Association (2000). Almost all technologies in the energy system are assigned exogenous costs and efficiencies (or, input-output coefficients); electric power plant costs and efficiencies are from the inputs to the 2016 Annual Energy Outlook (EIA, 2016), though historical efficiencies are calibrated based on the energy balance data. A similar approach is taken for other technologies.

Final demand sectors include buildings (residential and commercial), transportation (passenger and freight, including road, rail, air, and shipping), and industrial (fertilizer, cement, and general industry) sectors. The input data to the transportation sector is documented in Mishra et al. (2013), and the input data to the buildings sector is documented in Clarke et al. (2018). Fertilizer production assumptions come mostly from IEA (2007), and cement production assumptions are from Worrell et al. (2001) and IEA (2007). The energy module also includes simple representations of a number of urban processes, such as wastewater treatment, landfills, and industrial processes that generate emissions.

In GCAM v5.1 equipment vintages are explicitly accounted for in several sectors: electric generation, passenger cars and trucks, freight trucks, liquid refining plants, and fertilizer production. Older technologies operate as long as the price of the good produced exceeds the variable cost of operation. New technologies are always assumed to operate, but the decision to construct these technologies depends on both the variable cost and the investment cost.

The primary outputs of the energy system are: energy consumption by all sectors, energy production by the transformation sectors, energy prices, and emissions of $CO_2$ and other species. The final demands include passenger-kilometers travelled, freight tonne-kilometers shipped, buildings sector floorspace levels and service outputs, cement production and associated emissions and energy requirements, and fertilizer production volumes.

**2.7 Land**

The land component of GCAM calculates supply, demand, and land use for food, feed, fiber, forestry, and bioenergy products, as well as land cover for natural ecosystem types. GCAM includes all commodities reported by the FAO, but aggregates them into 15 commodity classes (e.g., Corn, Rice, Wheat, SugarCrop, OilCrop, Forest, Pasture, etc.). Demands for food, fiber, and forestry are driven by the size of the population, their income levels, and commodity prices. Food demand is price responsive,

but with relatively low price elasticities (-0.08 for crops and -0.25 for meat and dairy). Livestock can be pasture-fed or fed a mix of grains and pasture; future shares of each type depend on initial shares, the logit exponent and the costs of inputs. Feed demand is determined by the size of the livestock herd, the share of grain-fed animals, and the feed mix. Demand for commercial bioenergy is determined by the energy system, as described above, and includes primary (solid) biomass and secondary gases and liquids derived from biomass. Agricultural demands are modeled at the economic region level, with 32 regions globally represented in GCAM v5.1.

Supply for these products depends on the land allocated to that use and its yield. Land is allocated among a number of uses assuming that land owners maximize expected profit. However, GCAM uses a logit formulation, assuming that the cost of production is not identical across all producers. As a result, an increase in the profit rate for one type of land will result in an increase in the share of that land; however, all land is not typically allocated to the type with the highest profit rate, see Wise et al. (2014a) for more information. GCAM includes a comprehensive set of land use (e.g., crops, pasture, commercial forest, urban) and land cover types (e.g., grass, shrub, tundra, non-commercial forest, other arable land). Land allocation and agricultural supply are determined within each land use region, which is specified by a combination of economic region and water basin, i.e. GCAM v5.1 has 384 land use regions (see Figure 2).

GCAM v5.1 includes endogenous future agricultural yield changes, including the potential for price-induced intensification. The model includes four different technologies for each commodity within each region: irrigated/high fertilizer, irrigated/low fertilizer, rainfed/high fertilizer, and rainfed/low fertilizer, each with a different yield and cost of production. The share that each technology receives depends on the profitability of that technology. In general, increases in water costs will lead to higher shares of rainfed crops; increases in fertilizer costs will lead to higher shares of low-fertilizer technologies; and increases in land competition will result in movement toward higher yielding technologies.

Most agricultural products are traded on the global market using a net trade approach, in which global supply matches global demand at each time step. Bioenergy is modeled as a regional market, with the potential for trade. GCAM v5.1 uses a logit formulation to determine the share of domestic versus imported bioenergy consumed in each region, as well as the regions' contribution to the traded bioenergy market.

Agriculture and land use emissions are calculated at each time step. GCAM calculates land-use change $CO_2$ emissions using an accounting-style approach, similar to that of Houghton (1995). GCAM estimates the equilibrium change in carbon due to a change in land use/land cover and then allocates that change across time. The profile of emissions across time varies depending on whether the carbon is above or below ground, whether there is an increase in carbon or a decrease, and the user-specified time to maturity (i.e. peak standing carbon stock; slow-growing higher latitudes having longer times to maturity). For example, a decrease in forest cover will result in an immediate pulse of aboveground carbon to the atmosphere, while the carbon

sequestered as a result of an increase in forest will be spread over time. Non-$CO_2$ emissions depend on the level of activity, the initial emissions coefficient, and any emissions controls applied. For example, $CH_4$ emissions from livestock will increase as the livestock production increases but decline with a carbon price due to the imposition of a Marginal Abatement Cost (MAC) curve.

GCAM includes the ability to represent multiple different types of land-related policies, including afforestation, protected lands, bioenergy constraints (e.g., lower or upper bounds on total bioenergy consumption or the share of bioenergy in liquid fuels), and bioenergy taxes (Calvin et al., 2014). These options can be specified by region and time period. The default policy assumption is that 90% of natural ecosystems are protected.

There are three primary types of input data for the land component: historical data used for calibration, information related to competition, and future driver data. The historical data includes: supply, demand, prices, and variable costs of production for agriculture and forestry products; land use and land cover; the value of unmanaged land; carbon cycle parameters; and emissions and/or emissions factors for non-$CO_2$s. Supply, demand, and prices are derived from Food and Agriculture

Organization statistics (FAO, 2018). Production costs are obtained from the U.S. Department of Agriculture (USDA, 2018). Land use and land cover are derived from a variety of sources, as documented in Di Vittorio et al. (2016). The carbon cycle parameters include carbon densities for above and below ground stocks, and the number of years to maturity. Emissions and emissions factors for non-$CO_2$ gases are derived from EDGAR (JRC, 2011) for most gases and Bond et al. (2007) and Lamarque et al. (2011) for black carbon and organic carbon. The competition information includes the logit exponents dictating

the competition between various land types.

The future driver data includes income and price elasticities for demand; agricultural productivity growth rates; and MAC curves. Additionally, demand is driven by the income and population as described earlier. Income elasticities are derived from FAO estimates of future agricultural demand or are estimated to ensure demand is consistent with historical relationships

between income and caloric intake, depending on the scenario.

The primary outputs of the land component of GCAM are supply of agriculture and forestry products; demand for agriculture and forestry products; prices for agriculture and forestry products; land use and land cover by type; and agriculture, land use, and land use change emissions for all greenhouse gases, short-lived species, and ozone precursors.

**2.8 Water**

The water component of GCAM calculates water supply and demand for each region and sector within the model. In GCAM v5.1, water supply is an unlimited resource, including all sources of water (e.g., freshwater, groundwater, seawater). The price for this resource can be specified by the user. GCAM tracks water demand for irrigation (Kim et al., 2016), electricity

generation (Davies et al., 2013; Kyle et al., 2013; Liu et al., 2014), municipal uses (Hejazi et al., 2013), industrial manufacturing, primary energy production, and livestock (Hejazi et al., 2014a). For each type of water, GCAM tracks both withdrawals and consumption. In general, water withdrawal indicates the total water extracted from a water supply system, while water consumption indicates that the water is used by consumers in a way that it cannot be returned and reused immediately. Municipal water demands are driven by changes in population, per capita GDP, and technological change (Hejazi et al., 2013); all other water demands are modeled as inputs to otherwise existing technologies in the energy and land systems.

Water demand for irrigation depends on the share of irrigated land (see Section 2.6), other available water, and the water coefficient (water demand per unit output). Water coefficients vary by crop and region. In term of irrigation, water withdrawals refer to irrigation water applied to agricultural fields, including evapotranspiration requirements of crops that are met by irrigation water plus any field losses of water. Water consumption refers to the evapotranspiration requirements of the crops that is met by irrigation water. In addition to tracking withdrawals and consumption, GCAM also tracks biophysical water consumption for crops, which applies to both rainfed and irrigated technologies within any basin and crop type, is the sum of water consumption (as described above) and soil moisture from precipitation, used by plants via transpiration. Livestock water demand depends on a region-specific coefficient which represents both animal drinking water, plus any other water used by the animal production operations. The coefficients are in units of cubic meters of water per kilogram of animal commodity produced (e.g., beef, dairy, etc).

For electricity water demands (Davies et al., 2013; Kyle et al., 2013; Liu et al., 2014), GCAM has exogenously assigned water withdrawal and consumption coefficients for each region and generation technology based on Macknick et al. (2011). Cooling system options compete using the logit formulation described above. While the competition between cooling system options is endogenous and cost-based, because water prices are constant in GCAM v5.1, the model output tends to largely reflect the exogenous share-weight assumptions, which follow Davies et al. (2013). Specifically, most regions are assumed to shift from once-through to recirculating systems over time, but regions that primarily use seawater at present are assumed to continue to do so in all future time periods.

The industrial manufacturing sector's water demands scale with industrial output. Water demands for primary energy production depends on fuel production and the bottom-up estimates of water demand per unit energy produced for the following fuels: coal, oil (conventional and unconventional), natural gas, and uranium (Hejazi et al., 2014a).

Historical water withdrawal and consumption data are from multiple sources including FAO-AQUASTAT (FAO, 2016) and the USGS (USGS, 2016). The irrigation water demand estimates are derived from gridded and nation-level estimates of Mekonnen and Hoekstra (2011). The loss coefficients for conveyance and other field losses are from the country-level estimates according to Rohwer et al. (2007). The livestock coefficients are calculated from Mekonnen and Hoekstra (2010),

which provide total water demands in liters of water per animal per day, by country, for a base year of 2000. The water withdrawal and consumption coefficients for each region and electricity generation technology are from Macknick et al. (2011). The capital costs of different cooling technologies are obtained from National Energy Technology Lab (2008). Water consumption data for manufacturing is obtained from the Vassolo and Döll (2005) global inventory of manufacturing and electric power water demands for a base year of 1995, augmented with some additional data from Kenny et al. (2009). Municipal water prices obtained from the International Benchmarking Network for Water and Sanitation Utilities (IBNET, 2016), and overall municipal water supply efficiency are based on Shiklomanov (2000). Water demand for primary energy production is from Maheu (2009), augmented with some additional data from Kenny et al. (2009) and Solley et al. (1998).

The primary outputs of the water component of GCAM are water withdrawals and consumption for each region, sector, and technology. Additionally, GCAM computes biophysical water consumption for crop production.

## 2.9 Climate

GCAM v5.1 includes Hector v2.0, an open-source, object oriented, reduced form climate carbon-cycle model. Reduced-complexity or simple climate models represent the most critical global-scale earth system processes with low spatial and temporal resolution. Hector v2.0, like other simple climate models, calculates future concentrations of greenhouse gases from a given emissions pathway while modeling carbon and other gas cycles; calculates global mean radiative forcing from greenhouse gas concentrations and short-lived climate forcers; and converts the radiative forcing to global mean temperature and other Earth system variables (Hartin et al., 2015; Meinshausen et al., 2011).

Hector has a three-part carbon cycle: atmosphere, land and ocean. The atmosphere is treated as a single well-mixed box, where a change in atmospheric carbon is a function of anthropogenic fossil fuel and industrial emissions, land-use change emissions, the atmosphere-ocean and the atmosphere-land carbon fluxes. In Hector's default terrestrial carbon cycle, vegetation, detritus, and soil are linked with one another and to the atmosphere by first-order differential equations. Net primary production is a function of atmospheric $CO_2$ and temperature. Carbon flows from vegetation to the detritus and then down to soil, where some fraction is lost due to heterotrophic respiration. The terrestrial carbon balance at any time is the difference between net primary production (NPP) and heterotrophic respiration (RH) summed over the user-specified geographical regions (global in GCAM 5.1). NPP is modified by a user-defined carbon fertilization parameter. Changes in RH are controlled by a user-defined temperature sensitivity.

The surface ocean carbon flux is dependent upon the solubility of $CO_2$ within high and low latitude surface boxes which are calculated from an inorganic chemistry submodule (Hartin et al., 2016). Hector calculates $pCO_2$, pH and carbonate saturations in the surface boxes; once carbon enters the surface boxes, it is circulated through the intermediate and deep ocean layers via water mass advection and exchanges, simulating a simple thermohaline circulation.

Radiative forcing is calculated from each individual atmospheric constituent; $CO_2$, halocarbons, NMVOC, black carbon, organic carbon, sulfate aerosols, $CH_4$, and $N_2O$, along with forcing from tropospheric ozone and stratospheric water vapor. $CO_2$, $CH_4$, $N_2O$, and halocarbons are converted to concentrations, while NMVOC, and aerosols are left as emissions (Hartin et al., 2015).

Global atmospheric temperature is a function of user-specified climate feedback parameter, which indicates the equilibrium climate sensitivity for a doubling of $CO_2$, total radiative forcing and oceanic heat flux. Atmosphere-ocean heat exchange in Hector v2.0 consists of a one-dimensional diffusive heat and energy-balance model, DOECLIM (Kriegler, 2005). This is a significant improvement over Hector v1.1, better representing the ocean's mixed layer and deep ocean heat uptake, and resulting in an improved simulation of global mean temperature.

At every simulation time step, beginning in 2005 GCAM supplies Hector with global emissions of: fossil fuel and industrial $CO_2$, land-use change $CO_2$, $CH_4$, $N_2O$, BC, OC, CO, NMVOC, and halocarbons ($C_2F_6$, $CF_4$, $SF_6$, HFC134a, HFC32, HFC125, HFC227ea, HFC23, HFC134a, HFC245fa). The parameter values used in Hector are documented in Hartin et al. (2015).

The primary outputs of the climate component of GCAM v5.1 are global mean temperature, ocean heat uptake (both mixed layer and deep ocean), $CO_2$, $CH_4$, $N_2O$ and halocarbon concentrations, radiative forcing (both total and individual components), carbon fluxes both on land and ocean, and carbon cycle output (e.g., NPP, RH, ocean pH, carbonate saturations).

**2.10 Example of a Coupled System: Bioenergy**

The five GCAM components are linked in code, with different types of information exchanged among them depending on the component and the variable of interest. As an example, we describe GCAM v5.1's representation of bioenergy. Bioenergy demand is determined by the *energy* system and depends on the scale of the economy (determined by the *socioeconomic* system), the price of bioenergy, the capital and O&M costs of bioenergy technologies, as well as the cost and prices of competing energy technologies. Supply of bioenergy is determined largely by the *land* system and depends on the price and cost of bioenergy, as well as the price and cost of competing land types. The cost of bioenergy includes costs of fertilizer (produced by the *energy* system) and irrigation (supplied by the *water* system) if these management practices are used. The price of bioenergy is adjusted by the solution mechanism until supply and demand equilibrate to within the user-specified solution tolerance.

In the *land* system, the production of bioenergy can result in emissions of $CO_2$ due to land use change and non-$CO_2$ emissions (e.g., $N_2O$ emissions from fertilizer application). In the *energy* system, GCAM accounts for the uptake of carbon during the growth of bioenergy, as well as the release of carbon during combustion. Without the use of $CO_2$ capture and storage (CCS),

this combination of uptake and release results in net zero emissions. If a technology with CCS is used together with bioenergy, net negative emissions result. These emissions are passed to the *climate* system (summed with all other global $CO_2$ emissions), which calculates the effect of bioenergy on atmospheric $CO_2$ concentration, radiative forcing, and temperature rise.

GCAM v5.1 includes a limit on the amount of net negative emissions in any period that is linked to GDP (from the *socioeconomic* system). This limit is to reflect that once emissions are net negative a carbon tax generates an expense to the economy and not a revenue. Currently, GCAM assumes that each region will allocate no more than 1% of GDP to negative emissions. The cost (and thus deployment) of bioenergy is adjusted to ensure this limit is not exceeded.

Fertilizer is another example of a tightly coupled system, with its production determined by the *energy* system and consumption determined by the *land* system. Additionally, many other aspects of GCAM create direct or indirect linkages among sectors (e.g., water demand is linked to the energy and agricultural production, climate is linked to emissions produced by the energy and land systems).

## 3 Description of the Scenarios

The results presented in the following sections include six different socioeconomic pathways: a core scenario (CORE, described in Section 2.4) and the five SSPs, as described in Calvin et al. (Calvin et al., 2017).[3] We combine each socioeconomic pathway with two different climate policy assumptions, resulting in twelve possible scenarios (Table 1). However, GCAM cannot limit radiative forcing to 2.6 W/m² in the SSP3 pathway due to limited technology and incomplete control of agriculture and land use emissions (Calvin et al., 2017; Fujimori et al., 2017). Thus, we include eleven scenarios in this paper.

*Table 1: Scenarios included in this paper. Scenarios are categorized by their socioeconomic pathway (columns) and their mitigation policy (rows). Cell values are the names of the scenarios used in the rest of the paper. Note that the SSP3-2.6 is infeasible and thus omitted from the table and the rest of the paper.*

| | | Socioeconomic Pathway | | | | | |
|---|---|---|---|---|---|---|---|
| | | **CORE** | **SSP1** | **SSP2** | **SSP3** | **SSP4** | **SSP5** |
| **Climate Policy** | **No climate policy** | CORE | SSP1 | SSP2 | SSP3 | SSP4 | SSP5 |
| | **Radiative forcing limited to 2.6 W/m² in 2100** | CORE-26 | SSP1-26 | SSP2-26 | | SSP4-26 | SSP5-26 |

---

[3] Note that the CORE scenario is similar to the SSP2. However, these two scenarios differ slightly in near-term GDP as described in Section 2.4, as well as small differences in non-$CO_2$ emissions factors and CCS assumptions as described in Calvin et al. (2017).

## 4 Results

The output file produced by GCAM v5.1 is approximately 2 Gigabytes per scenario, with more than 38,000 output variables per region and time period. In this section, we show highlights of these results for each major area. We begin by examining results for the CORE and CORE-26 (Sections 4.1 to 4.5), before looking at results from the alternative socioeconomic pathways (Section 4.6).

### 4.1 Socioeconomics

In the CORE scenario, global population grows through 2070, peaking at 9.5 billion, before declining (Figure 4, left panel). Global GDP increases by a factor of 6 between 2010 and 2100, an average growth of 2% per year (Figure 4, right). In GCAM v5.1, GDP and population are unaffected by other model components; thus, these variables are identical in the CORE and CORE-26 scenarios.

### 4.2 Energy

In the CORE scenario, electricity generation (Figure 5, top) and primary energy use (Figure 5, bottom) continue to rise over the rest of this century, as increases in income and population drive increased demand for energy services (e.g., passenger kilometers travelled, building energy, etc.). Fossil fuels remain the dominant source of energy, accounting for 86% of total primary energy consumption (60% of electricity) in 2100. However, the use of non-biomass renewables for electricity generation continues to rise, growing from 20% in 2010 to 27% in 2100. Despite this increase, non-biomass renewables remain a very small fraction of global primary energy consumption throughout the century. The continued dependence on fossil fuels in the CORE scenario results in an increase in fossil fuel and industrial $CO_2$ emissions, which increase from 8.9 GtC/year in 2010 to 19.7 GtC/year in 2100.

Imposing a carbon price, as in the CORE-26, increases the cost of fossil fuel use, incentivizing substitution by lower carbon fuels. The result is an increase in electricity generation in total, as end users shift from direct consumption of oil and gas to electricity use. Additionally, the electricity generation mix shifts dramatically, with 11% of generation met by bioenergy with CCS, 26% from nuclear, and 43% by non-biomass renewables in 2100. Bioenergy's contribution to refined liquids also increases substantially in the CORE-26. As a result, bioenergy with CCS accounts for 40% of total primary energy consumption in 2100 (Figure 5, bottom right). The transition to low carbon (e.g., natural gas), no carbon (e.g., nuclear, renewables), and net negative carbon (e.g., bioenergy with CCS) fuels results in a substantial decrease in fossil fuel and industrial $CO_2$ emissions, with emissions in 2100 reaching -3 GtC/yr.

Energy consumption varies across region, in terms of both total consumption and fuel mix (Figure 6). Furthermore, these regional differences change over time due to differences in socioeconomic growth across regions, so the largest consumers today are not the largest consumers in the future. For example, the USA and China have the highest primary energy consumption in 2010, with 86 and 102 EJ/yr, respectively. In 2100, however, India and Africa_Western have the highest in

energy consumption in both the CORE (164 and 127 EJ/yr, respectively) and CORE-26 scenarios (75 and 86 EJ/yr, respectively). For fuel mix, there are regional differences in the share of fossil fuels used in 2010, with much lower shares in Africa_Western and Africa_Eastern than the rest of the world. However, in 2100, the biggest differences are across scenarios and not across regions, with fossil fuel consumption ranging from 70-95% of total primary energy in the CORE scenario and much lower use in the CORE-26.

**4.3 Land**

In the CORE scenario, income and population growth in the first half of the century result in increasing demand for agricultural products through 2050 (Figure 7, top left). Increases in agricultural productivity throughout the century balance these increases in production, resulting in nearly constant cropland area through 2050 (Figure 7, bottom left). Post-2050 the projected population declines together with continued yield improvements result in decreases in total agricultural production and

cropland area. The CORE scenario has modest demand for bioenergy (Figure 5), resulting in a small amount of land devoted to its production. Total agricultural area (crops, biomass, pasture) increases slightly throughout the century at the expense of natural ecosystems (forest, grass, and shrub).

In the CORE-26 scenario, the imposition of a carbon price incentivizes low carbon fuels in the energy system, resulting in

substantial increases in bioenergy demand. This results in a large expansion of bioenergy land, with ~7% of land devoted to bioenergy production in 2100 (Figure 7, bottom right). Increased competition for land with bioenergy results in increased food prices and consequently reduced demand (Figure 7, top right). Total agricultural area (crops, biomass, pasture) increases by 7.5% between 2010 and 2100 in the CORE-26 scenario, resulting in a decline in the extent of natural ecosystems. For example, forest cover decreases by 0.9 million km$^2$ (3%) between 2010 and 2100.

There are significant differences in land use across regions (Figure 8). However, regions that have large shares of cropland today (e.g., India, Europe, China, USA MidWest, Argentina) also have large shares of cropland in the future in both the CORE and CORE-26 scenarios. In the CORE scenario, bioenergy land is spread throughout the world's agricultural producing regions with only 16 of the 384 regions in GCAM devoting more than 10% of their land to bioenergy and only 1 very small region in

Southeast Asia devoting more than 20%. In the CORE-26, higher amounts of bioenergy land are required, resulting in shares of bioenergy land ranging from 0% to 58%. Note that some of the regions with large shares of bioenergy land are small in size. The largest amounts of bioenergy land in absolute value are in the Nile River basin in Africa_Eastern and the Niger River basin in Africa_Western, with 470 and 459 thous km$^2$ of bioenergy land in 2100 in the CORE-26, respectively. Only 10

region/basin combinations have more than 150 thous km$^2$ of bioenergy; these region/basins are found in Africa_Eastern, Africa_Western, India, Canada, and Russia.

For non-bioenergy crops, irrigation shares remain relatively constant over time, with approximately 20% of crops irrigated globally in both the CORE and CORE-26. Bioenergy crops are predominantly rainfed, with only 3-6% of these crops using irrigation. The use of the high fertilizer technology increases over the century, particularly in the CORE-26 scenario, rising from 50% to 56% between 2010 and 2100. As a result of both exogenous and endogenous yield growth, global average yields increase by 58% in the CORE and double in the CORE-26 between 2010 and 2100.

## 4.4 Water

In the CORE scenario, increases in demands for energy and agriculture result in increasing water consumption (Figure 9, top left) and withdrawal (Figure 9, bottom left) across all water sectors, with both consumption and withdrawal roughly doubling by the end of this century. The irrigation sector dominates water withdrawals and consumption throughout the century, with a much larger share of consumptive use. Industrial (manufacturing and electricity) and municipal water use are the next largest users of water, while livestock and primary energy production account for only 1-2% of total water use.

In the CORE-26 scenario, the imposition of a carbon price incentivizes low carbon fuels in the energy system. As a result, water use for electricity generation increases significantly under the CORE-26 scenario, mainly due to the large increase in water intensive technologies such as CCS (see Figure 5) and increased demand for electricity (Figure 5, top row). The CORE-26 also leads to large increases in bioenergy use; however, water use for bioenergy remains a small part of the overall total due to the dependence on rainfed bioenergy as described previously. Overall, climate policy results in a 14% increase in water consumption and a 17% increase in water withdrawal relative to the CORE scenario in 2100.

Water withdrawals differ significantly across region (Figure 10). The basins with the largest irrigation water withdrawals in 2010 are the Ganges, the Indus, and the Sabarmati. In 2100, the largest irrigation water withdrawals come from these three basins plus the Nile River basin (in both the CORE and CORE-26) and the Arabian Peninsula (in the CORE-26 only). The two largest regions in terms of non-irrigation water withdrawals are the USA and China in 2010 and India and China in 2100 in both the CORE and CORE-26 scenarios.

## 4.5 Climate

Absent any effort to mitigate (i.e., the CORE scenario), emissions of GHGs continue to rise throughout the century. Additionally, increases in pollution controls induced by rising incomes result in reduced emissions of sulfur and other aerosols. These increases lead to a rise in GHG concentrations, total radiative forcing, and global mean temperature. In particular, $CO_2$ concentration exceeds 700 ppmv and total radiative forcing exceeds 6 W/m$^2$ in 2100 (Figure 11).

In the CORE-26, a carbon price is applied to constrain the radiative forcing limit to 2.6 W/m$^2$ in 2100. This results in substantial reductions in GHG emissions. In this scenario, $CO_2$ concentration peaks around 450 ppmv mid-century, before declining to 400 ppmv (Figure 11, left panel). Total radiative forcing peaks around the same time at approximately 3.5 W/m$^2$ before declining to 2.6 W/m$^2$ (Figure 11, right panel) by the end of this century.

## 4.6 Alternative Socioeconomic Pathways

In addition to the default socioeconomic scenario (CORE), GCAM v5.1 includes five alternative scenarios, based on the SSPs (Riahi et al., 2017a). These scenarios span a range of challenges to mitigation and challenges to adaptation, with SSP1 having the lowest challenges, SSP2 having medium challenges, and SSP3 having high challenges. SSP4 has low challenges to mitigation, but high challenges to adaptation. SSP5 has high challenges to mitigation, but low challenges to adaptation. Storylines for all five SSPs are articulated in O'Neill et al. (2017).

The quantifications of population (Figure 12, left panel) and GDP (Figure 12, right panel) were developed by KC and Lutz (2017) and Dellink et al. (2017). Global population in 2100 varies across the five SSPs, with SSP1 and SSP5 having relatively low population (<7 billion) and SSP3 having high population (>12 billion). GDP, and GDP per capita, are highest in the SSP5, with total global GDP exceeding 700 trillion 2005$ per year. Despite its high population, SSP3 has the lowest GDP due to stagnant growth in income.

In addition to population and GDP, the GCAM implementation of the SSPs includes changes in the cost and performance of different technologies, as well as differences in effectiveness of air pollution policy and in the implementation of climate policy. These assumptions are documented in detail in Calvin et al. (2017). These changes result in a wide range of potential future pathways for energy (Figure 13), land (Figure 14), water, and climate. For example, primary energy use is highest in the SSP5, which has a high GDP. In addition, inexpensive fossil fuels in this scenario result in a continued dependence on coal, gas, and oil, which account for more than 90% of energy consumption in the SSP5 in 2100. In contrast, the SSP1 has low energy consumption and an increased dependence on renewables, due to its focus on sustainability (including both energy efficiency gains and low carbon fuel preferences).

If radiative forcing is limited to 2.6 W/m$^2$ by 2100, all scenarios transition towards low carbon fuels, with increased use of bioenergy with CCS (BECCS) in all cases. However, deployment of BECCS varies across scenarios, ranging from 200 EJ/year in the SSP2-26 to 323 EJ/year in the SSP5-26. Fossil fuel use in 2100 in these scenarios ranges from 35% in SSP1-26 to 53% in SSP5-26, as SSP5-26 has significant use of CCS. SSP1-26 has the highest deployment of non-biomass renewables, with 11% of total primary energy in 2100.

Land use and land cover also differ significantly across SSPs. The combination of high population and low agricultural productivity growth results in a large expansion of cropland area in the SSP3 (Figure 14). In contrast, SSP1 has a small population, with lower preferences for ruminant meat, and high agricultural productivity. As a result, cropland contracts in this scenario. Under the 2.6 W/m² policy, all scenarios show increases in land area devoted to bioenergy, but this trend is most prominent in the SSP4-26 and SSP5-26 due to the high demand for bioenergy with CCS described above. The SSPs also show increases in forest cover in the 2.6 W/m² scenarios due to the imposition of an afforestation incentive as part of the policy environment (Calvin et al., 2017; Kriegler et al., 2014). In the SSP1, SSP2, and SSP5, this policy is globally applied, resulting in increased forest cover in all regions. In the SSP4, afforestation is concentrated in middle and high income regions. This incentive is not included in the CORE-26; as a result, forest cover in the CORE-26 declines.

## 4.7 Comparison to Historical Data and Other Future Scenarios

This section compares GCAM results to both historical inventory data and other future projections. We include inventory data for energy, land, and $CO_2$ emissions. Energy data is from the International Energy Agency (IEA); in particular, we use total production of coal, gas, and oil from the IEA energy balances. Land data is from the Food and Agricultural Organization (FAO); in particular, we use cropland, pasture land, and forest land from FAOSTAT (FAO, 2018). $CO_2$ emissions are from the Carbon Dioxide Information Analysis Center (CDIAC; https://cdiac.ess-dive.lbl.gov/); in particular, we use total fossil fuel and industrial $CO_2$ emissions. GCAM uses IEA energy consumption and CDIAC $CO_2$ emission as inputs to the model; however, in the comparison presented, we use raw data from the various sources for comparison. For land cover, GCAM does not currently use FAOSTAT land cover variables.

For future projections, we compare GCAM results to the IPCC AR5 scenario database, described in Clarke et al. (2014), and the SSP scenario database, described in Riahi et al. (2017b). These databases have a large number of outputs from numerous scenarios and models (e.g., the AR5 database has ~1000 scenarios). We use all scenarios that include outputs for the full time period of interest (2010-2100), but focus on a small number of variables.

## 4.7.1 Historical Data

GCAM is initialized in the historical period to a variety of different datasets, depending on the variable of interest (see Section 2). Most of these variables are calibrated; that is, GCAM ensures that its model outputs exactly match the read-in observations. For example, GCAM reads in electricity generation by region, fuel types, and technology options. The calibration routines estimate the unobserved parameters (share weights) to ensure that the calculated values match the read in values. To test the validity of the calibration routine, Figure 15 compares GCAM outputs (x-axis) to observational data (y-axis) for global fossil fuel consumption (top left), global land cover (top right), and global $CO_2$ emission (bottom left). As presented in this figure, GCAM matches these data sources almost exactly, for the calibration period. Note that the biggest differences between GCAM

and observational data are for forest area. GCAM uses information from Meiyappan and Jain (2012), while the observational data shown in Figure 15 is from FAOSTAT.

### 4.7.3 Other Future Scenarios

In addition to comparing GCAM results to historical data, we also compare GCAM results to other scenarios in the literature, including both the database generated for the IPCC's 5[th] Assessment Report (Figure 16) and the SSP database (Figure 17).

In general, the eleven GCAM scenarios span the range of results presented in the literature, with the CORE scenario falling near the median. For example, primary energy consumption in 2100 in the GCAM v5.1 scenarios ranges from 735 to 1500 EJ per year without climate policy and from 480 to 900 EJ per year in the 2.6 W/m$^2$ scenarios, depending on the socioeconomic pathway (Figure 16, top left). In contrast, primary energy in the AR5 database ranges from 750 to 1850 EJ per year for the no climate policy scenarios and between 260 and 1000 EJ per year for the 2.6 W/m$^2$ scenarios. For fossil fuel and industrial $CO_2$ emissions (Figure 16, bottom left), the GCAM v5.1 no climate policy scenarios span a smaller range than those in the AR5, with 2100 emissions in GCAM v5.1 ranging from 10,600 MtC per year to 25,700 MtC per year. The AR5 database, in contrast, had a range of -2000 to 47,000 MtC per year. There is a similar difference in the range of total radiative forcing between GCAM v5.1 and AR5 (Figure 16, bottom right). For the 2.6 W/m$^2$ scenarios, GCAM v5.1 tends to overshoot more in the near-term (see Figure 16, bottom panels) than the AR5 scenarios. Some of these differences have to do with the timing of the climate policy; some of the AR5 scenarios, e.g., those from Clarke et al. (2009), had climate policy beginning in 2010. Other differences have to do with the availability and deployment of net negative emissions technologies, like BECCS.

Since the GCAM v5.1 scenarios include replications of the SSPs, we also compare our results to the SSPs presented in Riahi et al. (2017a). In this comparison, we can match specific socioeconomic scenarios and climate policies (Figure 17). The largest difference between GCAM v5.1 and the originally published GCAM SSP scenarios is for SSP3 and SSP5. In each of these scenarios, GCAM v5.1 uses less total primary energy (Figure 17, top), leading to lower fossil fuel and industrial $CO_2$ emissions (Figure 17, 2[nd] from bottom), and lower radiative forcing in the no climate policy scenarios (Figure 17, bottom). In each case, the GCAM v5.1 results are 20-25% below the median value from the SSPs. The difference in energy use and fuel mix between the GCAM4 results from the originally published GCAM SSPs and those presented in this paper are primarily due to updates in technology cost, as documented in Muratori et al. (2017).

### 5 Discussion and Conclusions

GCAM and similar models attempt to integrate a large set of human and Earth system dynamics and interactions taking place over many decades in the future into flexible and computationally tractable platforms. To date, the scientific capabilities embodied in GCAM and similar models have been important for informing both our scientific understanding of these

interactions and the decisions taken to better manage these systems. GCAM v5.1 describes a new version of GCAM, including several major enhancements from previous versions (e.g., water demand, multiple agricultural management practices, new land regions, new data system, newer climate model, alternative socioeconomic pathways).

At the same time, there is also a large set of dynamics and interactions that are not included in GCAM. For example, the version of GCAM presented in this paper does not include feedbacks from the global or regional climate to key systems such as energy (e.g., altering wind and solar power, air conditioning), water (altering water supplies, droughts), agriculture (altering crop yields through changes in temperature, precipitation, and growing seasons), among others. Similarly, GCAM v5.1 does not include dynamics at subregional scales such as counties or cities, although versions of GCAM with sub-regional detail

have been produced. Across the modeling community, groups are attempting to address these issues, adding more and more scope and complexity to their models. Of particular importance in GCAM development is the effect of a changing Earth system, and climate in particular, on energy, water, land, and economic systems. Research versions of GCAM already include new dynamics such as the effects of climate on water supplies (Hejazi et al., 2014b), energy demands (Clarke et al., 2018), and crop yields (Calvin and Fisher-Vanden, 2017; Kyle et al., 2014).

But these increases in scope and complexity raise challenges. With finer resolution and greater scope has come an increase in the computational demands of models like GCAM, including data storage needs. GCAM is increasingly being used on high-end computing clusters or other platforms with greater computational power and storage capabilities, which risks making it less useful to users without access to such high-end computational platforms. This is pushing towards a great focus in GCAM

on computational efficiency and on data management.

Finally, whereas all dynamics were previously included in the GCAM core, GCAM development is increasingly focused on creating an ecosystem of submodels that are designed to operate with GCAM and can be coupled in code. This includes, for example, a range of tools for providing information at spatial scales finer than those in the GCAM core. These options can be

included or not included depending on scientific questions, the modeling needs and computational capabilities.

## 6 Code Availability

GCAM is an open source model. The version of GCAM described in this paper is archived on both GitHub and Zenodo (doi:10.5281/zenodo.1308172). All code and inputs are available at: https://github.com/JGCRI/gcam-core. A user guide for GCAM is available at: http://jgcri.github.io/gcam-doc/user-guide.html. The GCAM development team hosts annual trainings

for GCAM (see http://www.globalchange.umd.edu/ for more information).

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

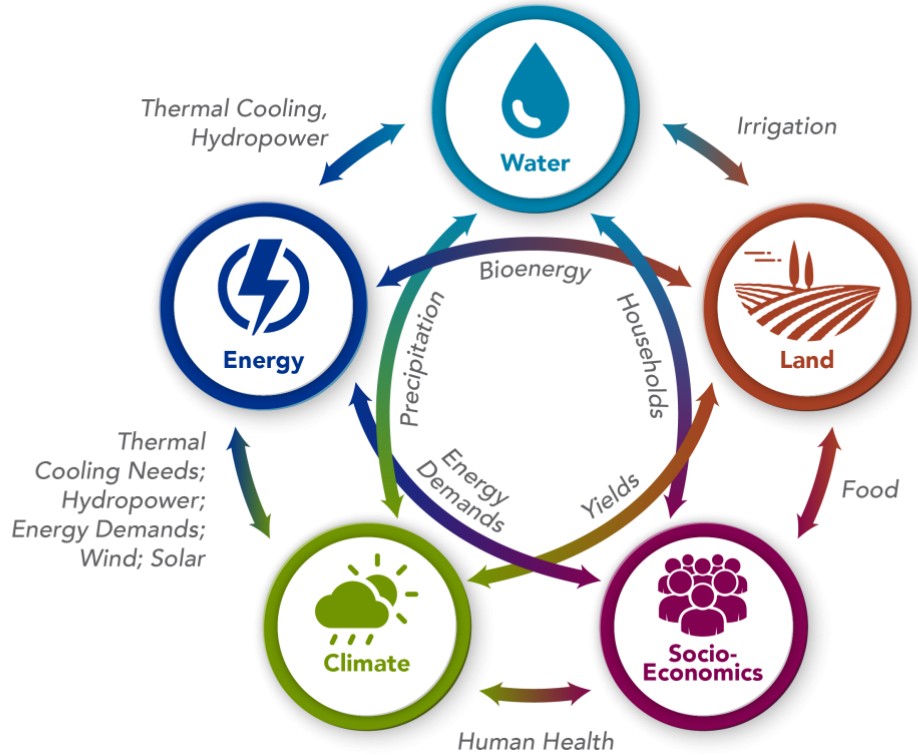

**Figure 1: Linkages between the five major systems (energy, water, land, socioeconomics, climate) in GCAM v5.1**

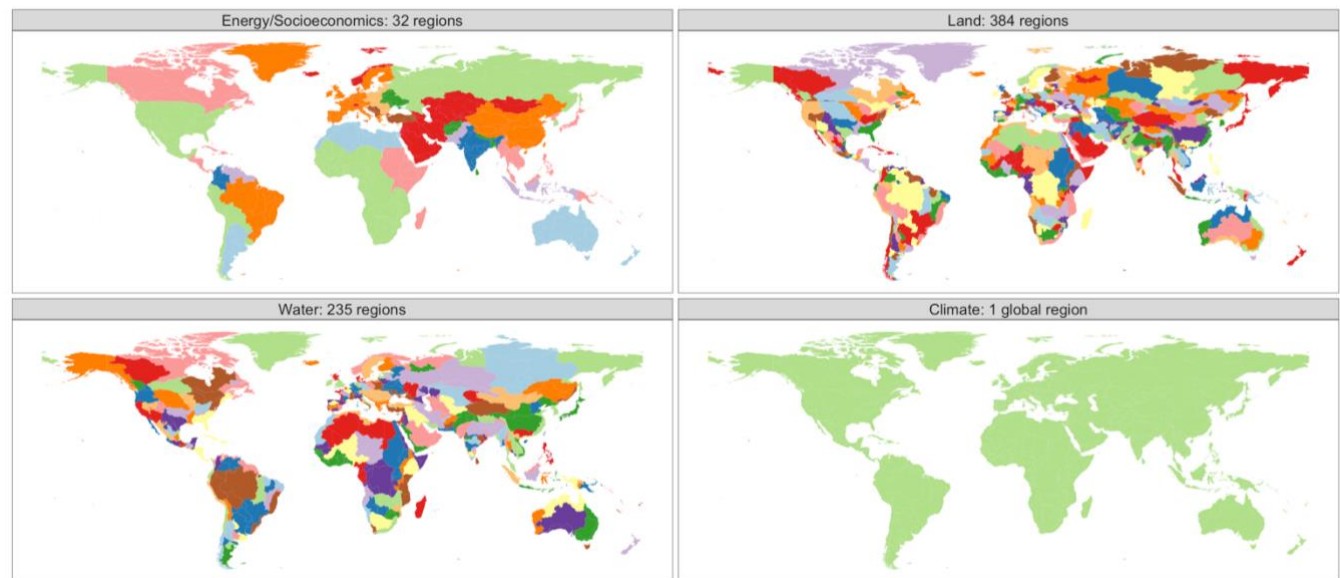

**Figure 2: GCAM regional mapping for energy & economy (top left), land (top right), water (bottom left), and climate (bottom right).** Regions are based on geopolitical boundaries for energy and economy, on water basins for water, and on a combination of geopolitical boundaries and water basins for land.

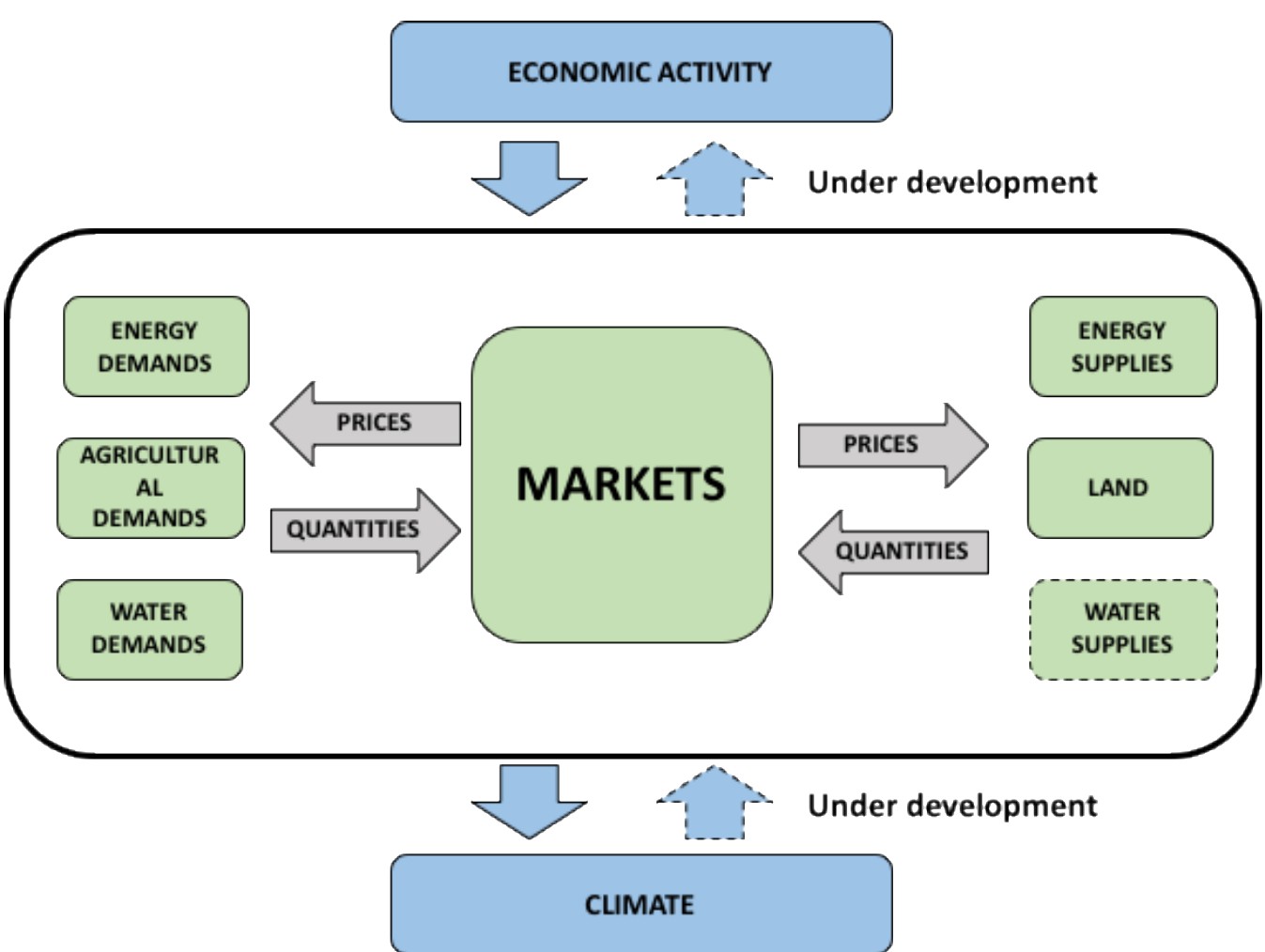

**Figure 3: Conceptual Schematic of the Operation of the GCAM Core.**

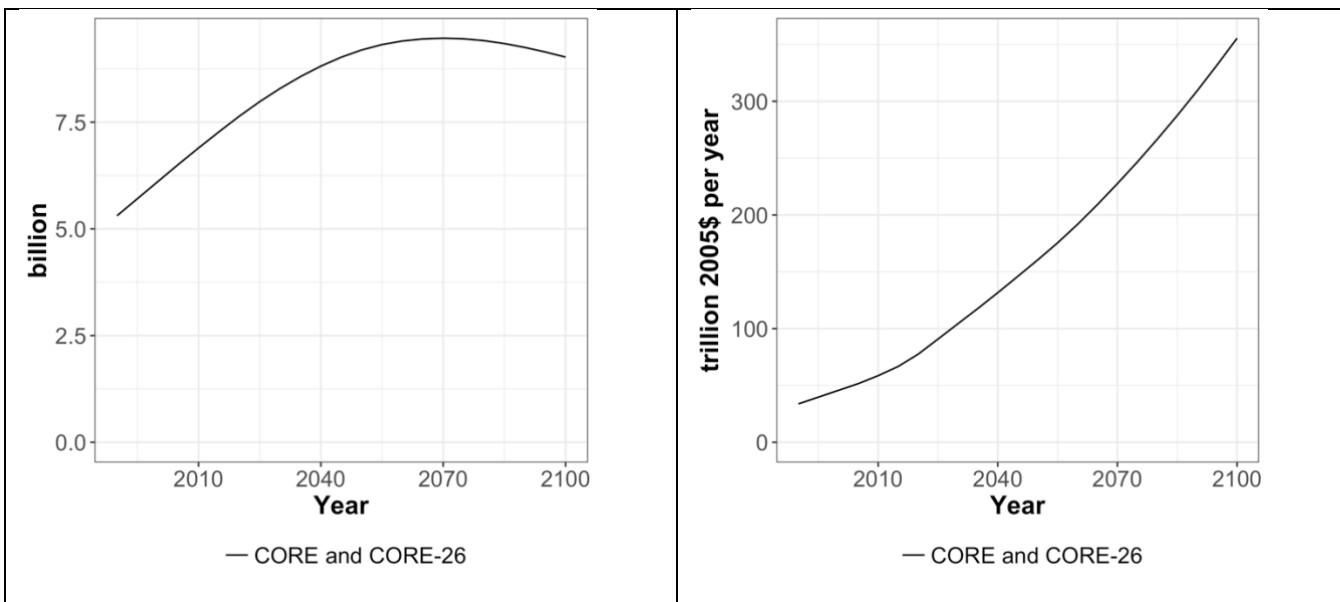

**Figure 4: Global population (left) and Gross Domestic Product (GDP) (right) for the CORE and CORE-26 scenarios.** GDP is reported in constant U.S. dollars (2005$), using market exchange rates. Note that both population and GDP are exogenous in GCAM v5.1, and thus, do not change with mitigation policy or any other factor.

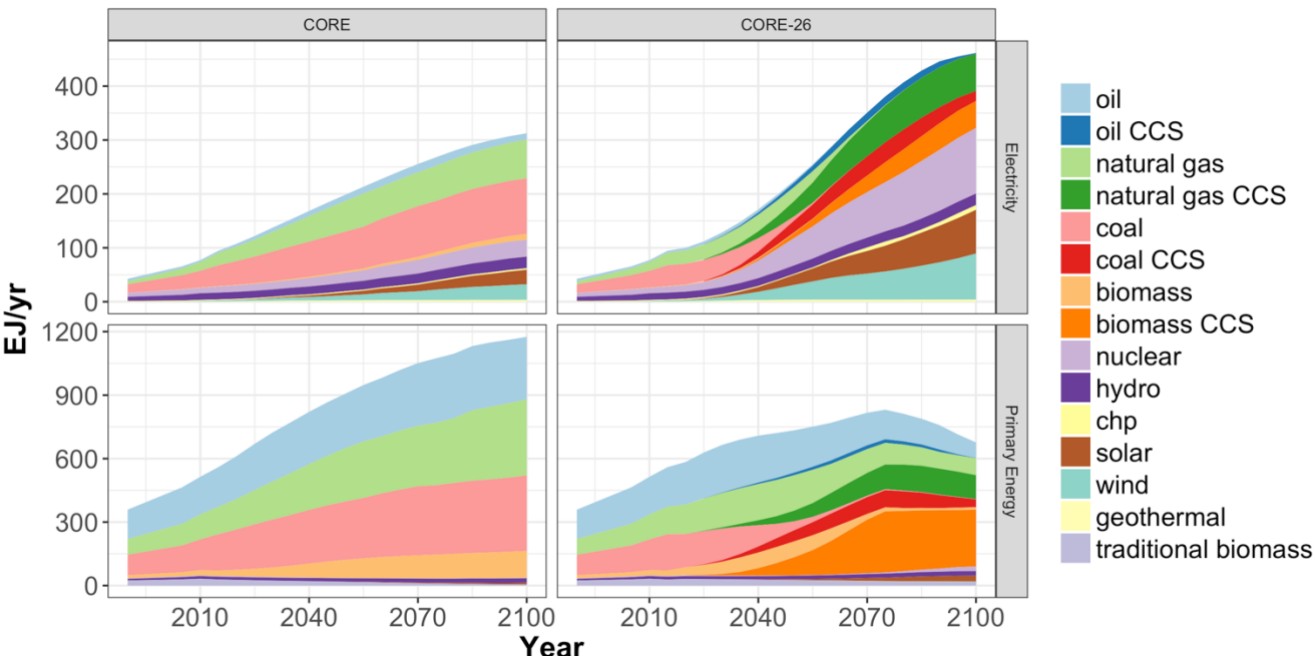

**Figure 5: Global electricity generation (top) and global primary energy consumption (bottom) for the CORE (left) and CORE-26 (right) scenarios.** Primary energy is reported using direct equivalent; that is, 1 EJ of nuclear or renewable electricity is reported as 1 EJ of primary energy consumption.

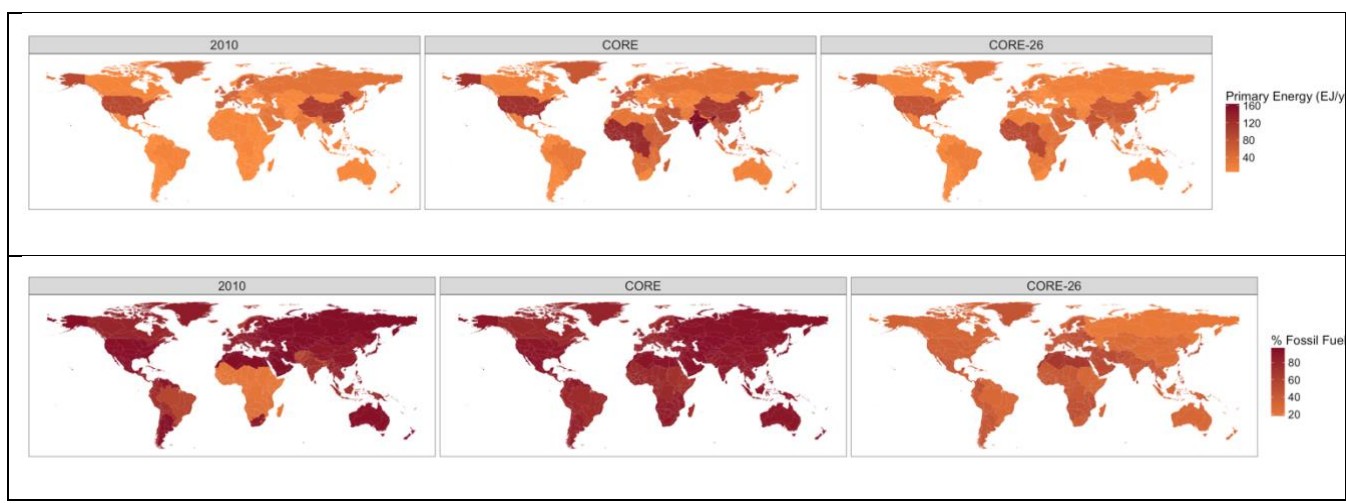

5    **Figure 6: Regional primary energy consumption (top) and % of primary energy from fossil fuels (bottom) in 2010 (left) and 2100 in the CORE (middle) and CORE-26 (right) scenarios.** Primary energy is reported using direct equivalent; that is, 1 EJ of nuclear or renewable electricity is reported as 1 EJ of primary energy consumption.

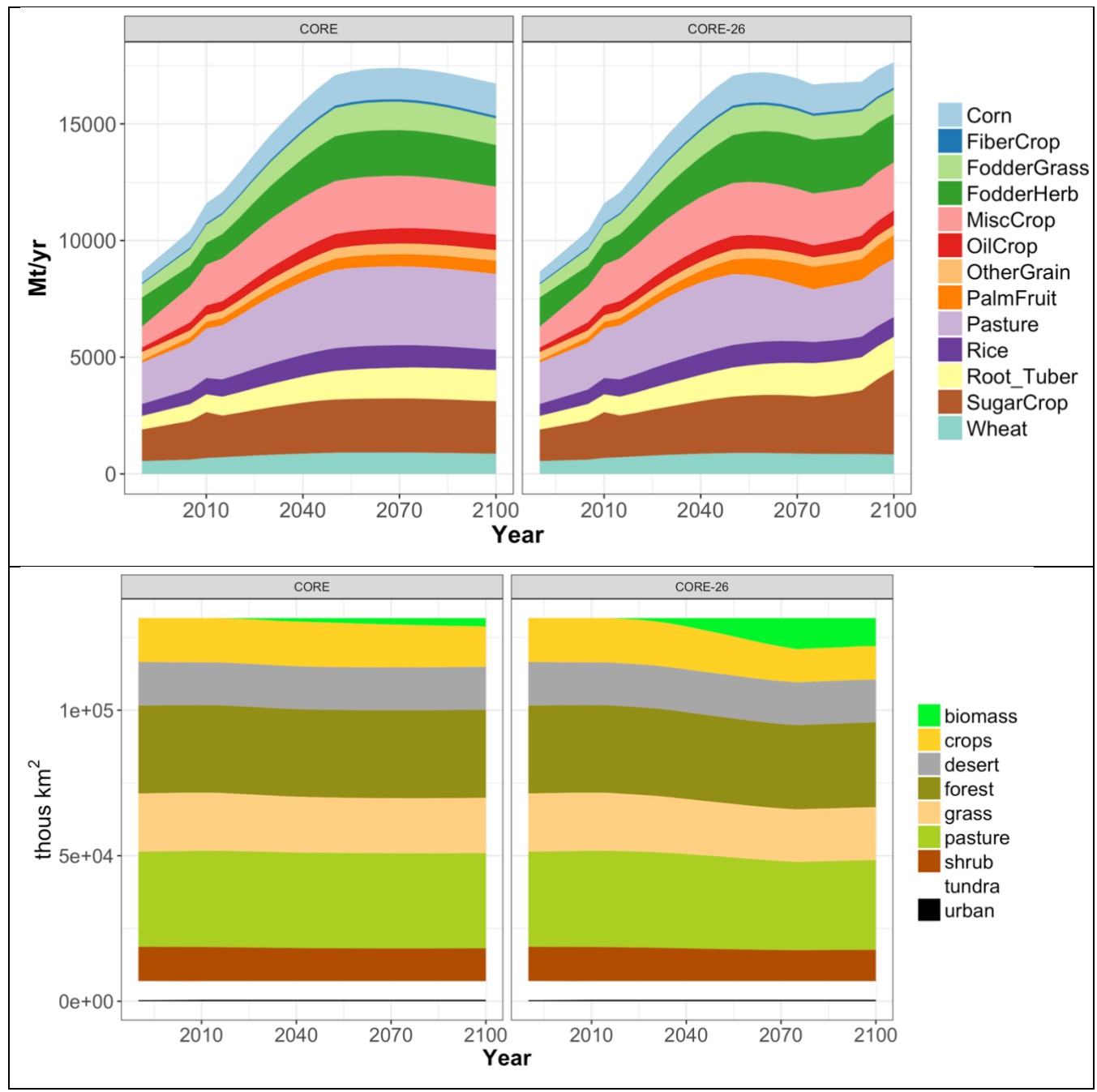

**Figure 7: Global agricultural production (top) and global land allocation (bottom) for the CORE (left) and CORE-26 (right).** Note bioenergy and forest are excluded from agricultural production (top) as they are modeled in different units, EJ/yr and m³/yr, respectively. Land cover data (bottom) are aggregated from the more detailed categories included in GCAM v5.1 for purposes of plotting.

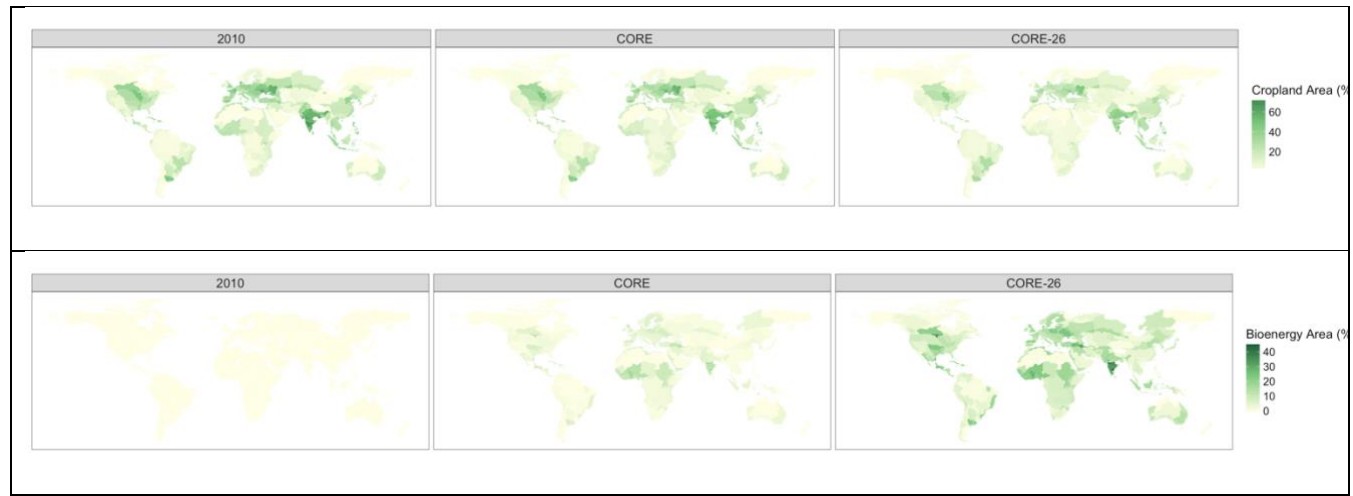

**Figure 8: Percentage of regional land area devoted to cropland (top) and bioenergy (bottom) in 2010 (left) and 2100 in the CORE (middle) and CORE-26 (right) scenarios.** Note that there is no dedicated bioenergy cropland in 2010 in GCAM; hence, the lower left map has zero values everywhere.

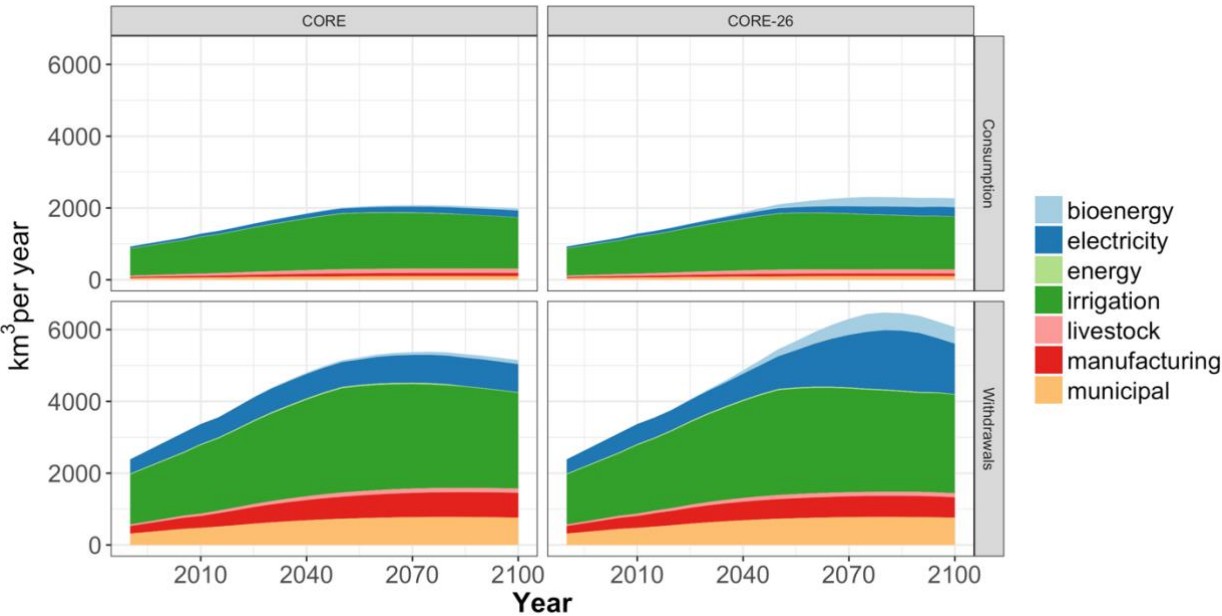

5    **Figure 9: Global water consumption (top) and global water withdrawals (bottom) by sector for the CORE (left) and CORE-26 (right) scenarios.** Data are aggregated from technology to sector for purposes of plotting.

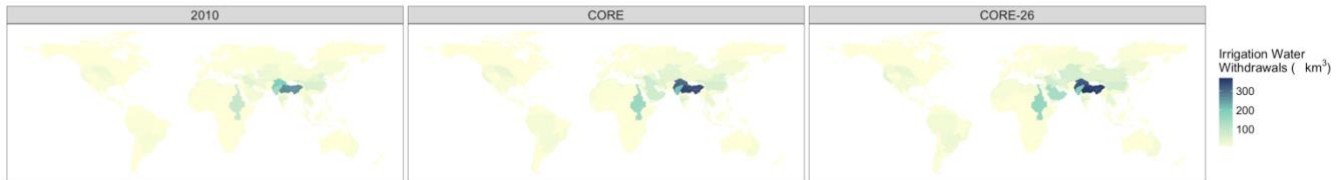

**Figure 10: Regional water withdrawals for irrigation in 2010 (left) and 2100 for the CORE (middle) and CORE-26 (right) scenarios.** Figure shows total irrigation, including both bioenergy and crops.

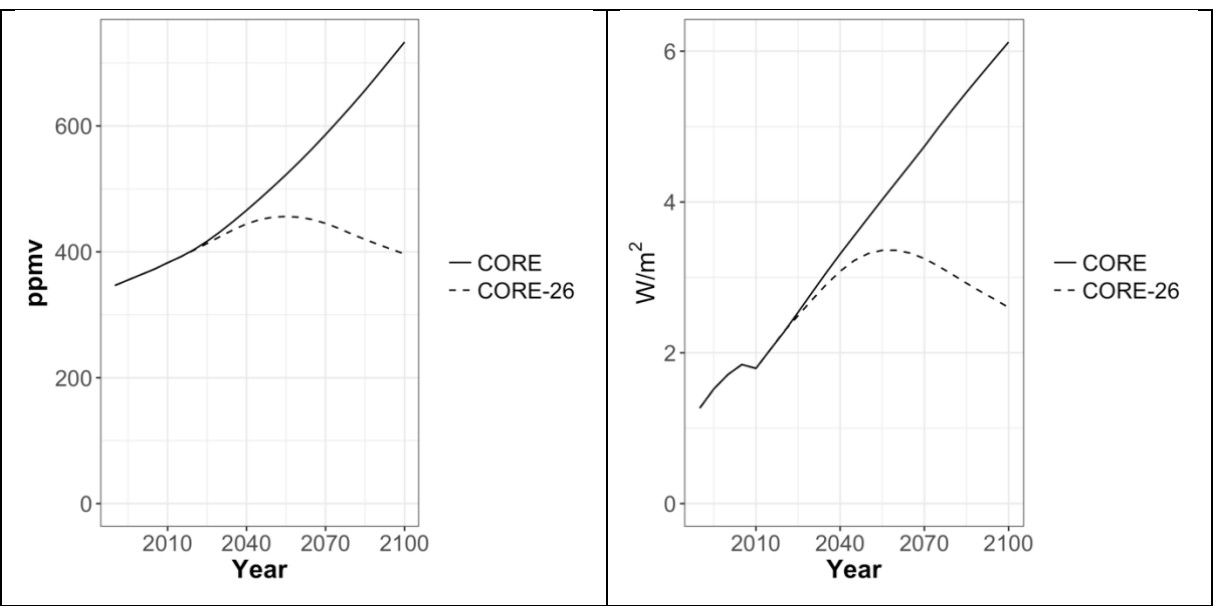

5    **Figure 11: CO₂ concentrations (left) and total radiative forcing (right) for the CORE (solid) and CORE-26 (dashed) scenarios.**

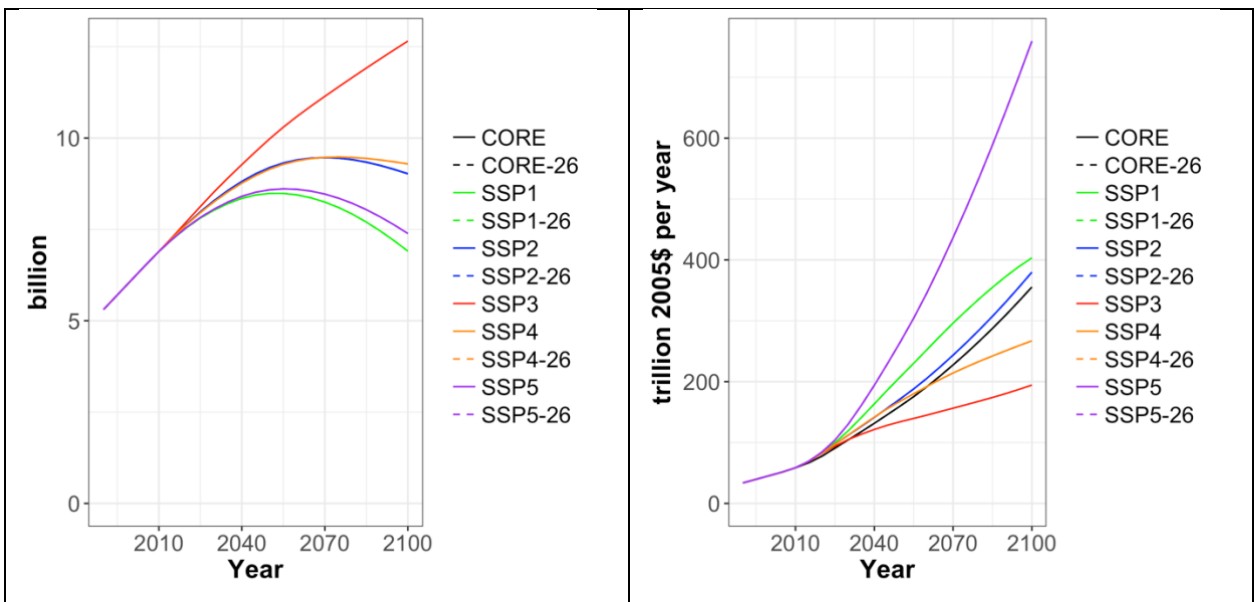

**Figure 12: Global Population and GDP for all scenarios.** The CORE scenario is virtually identical to the SSP2, but does differ in near-term GDP. Note that GCAM v5.1 does not model the effects of climate policy on either GDP or population.

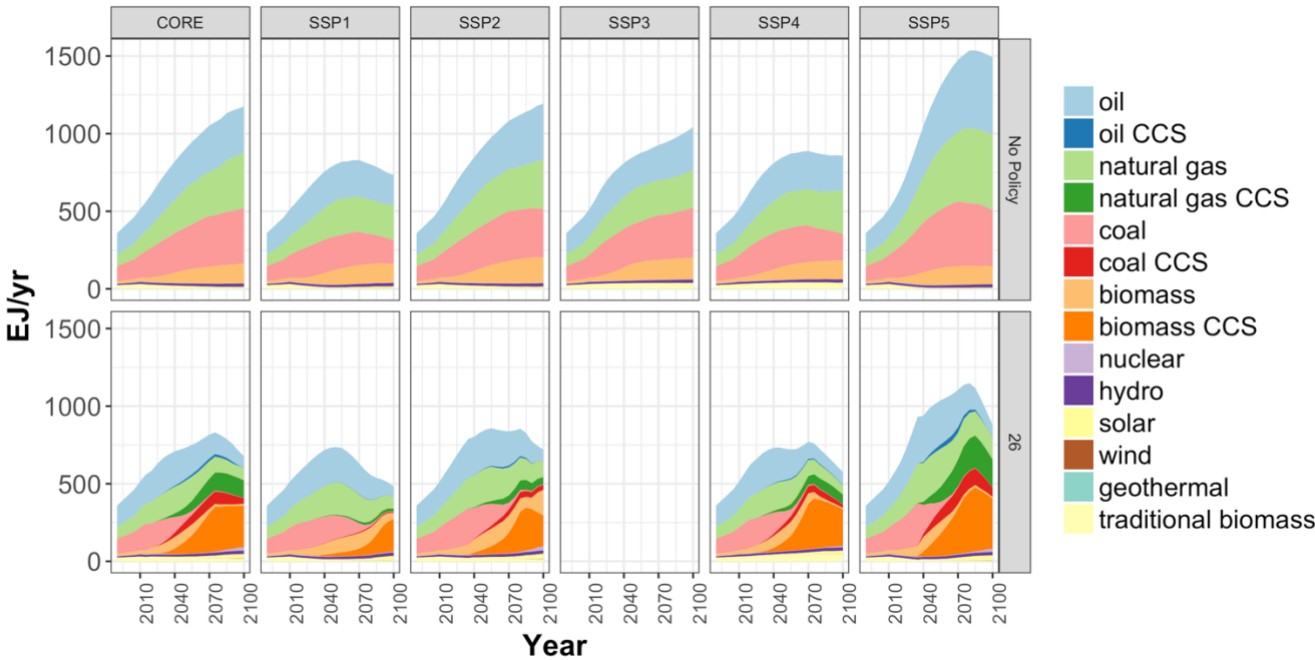

5   **Figure 13: Global Primary Energy by Fuel for all Scenarios.** Top row is without climate policy; bottom row are the 2.6 W/m² scenarios. Columns indicate the underlying socioeconomic assumptions.

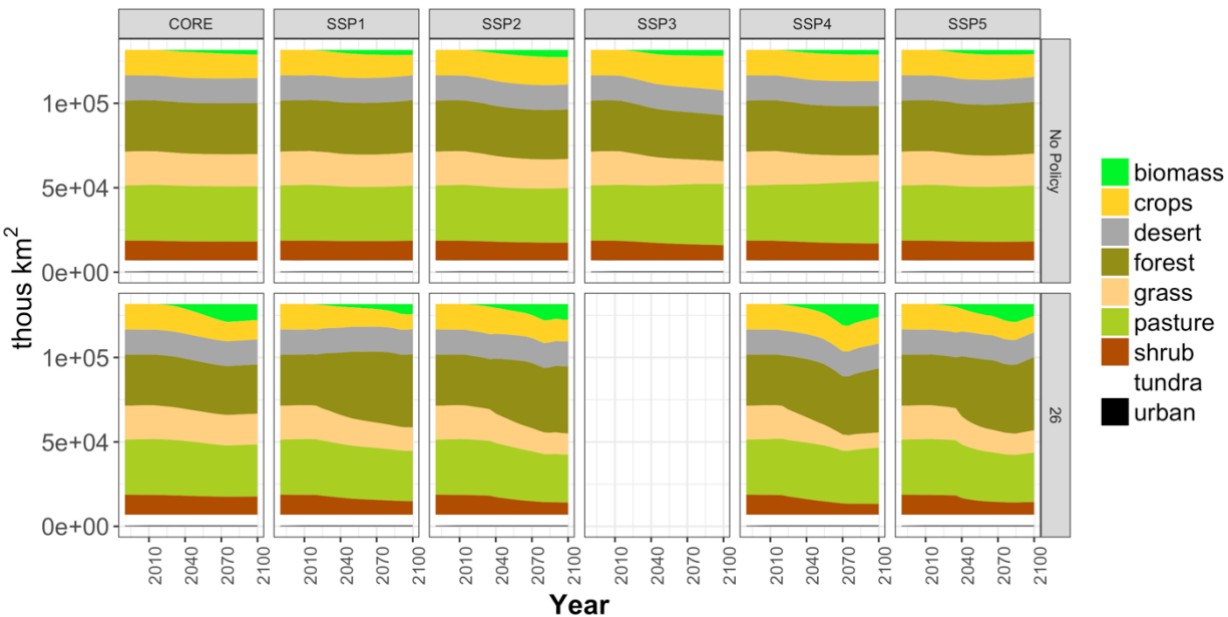

**Figure 14: Global land cover by type for all scenarios.** Top row is without climate policy; bottom row are the 2.6 W/m² scenarios. Columns indicate the underlying socioeconomic assumptions.

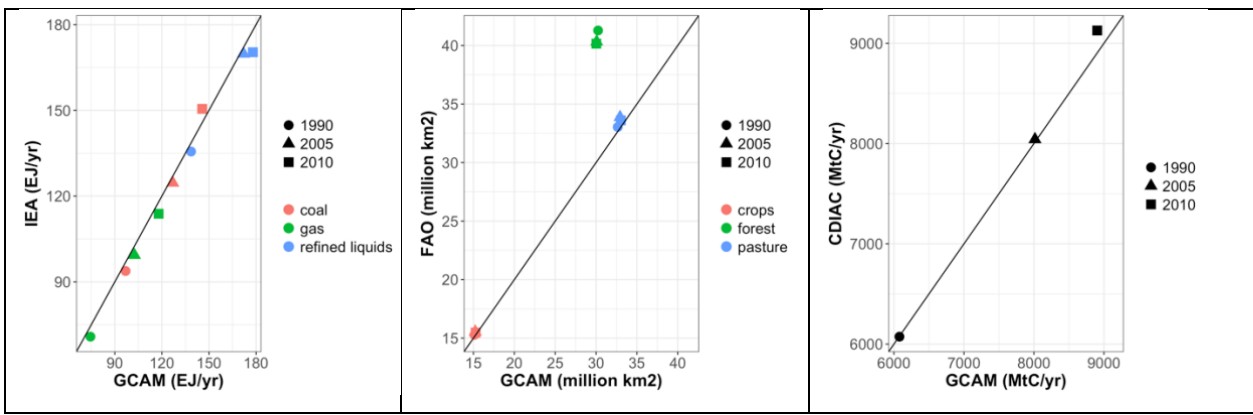

**Figure 15: GCAM results versus historical inventory data.** Left is primary energy consumption for fossil fuels, compared to IEA. Middle is global land cover by type, compared to FAO. Right is global fossil fuel and industrial $CO_2$ emissions, compared to CDIAC. Data is from the CORE scenario, but all GCAM scenarios are identical in the historical period.

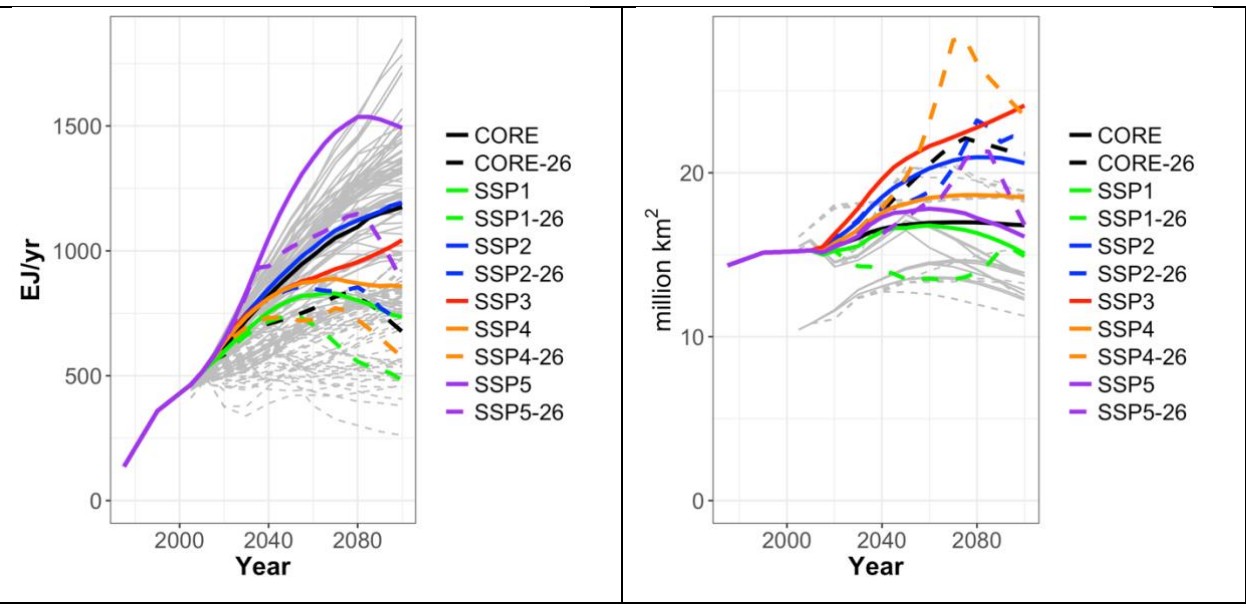

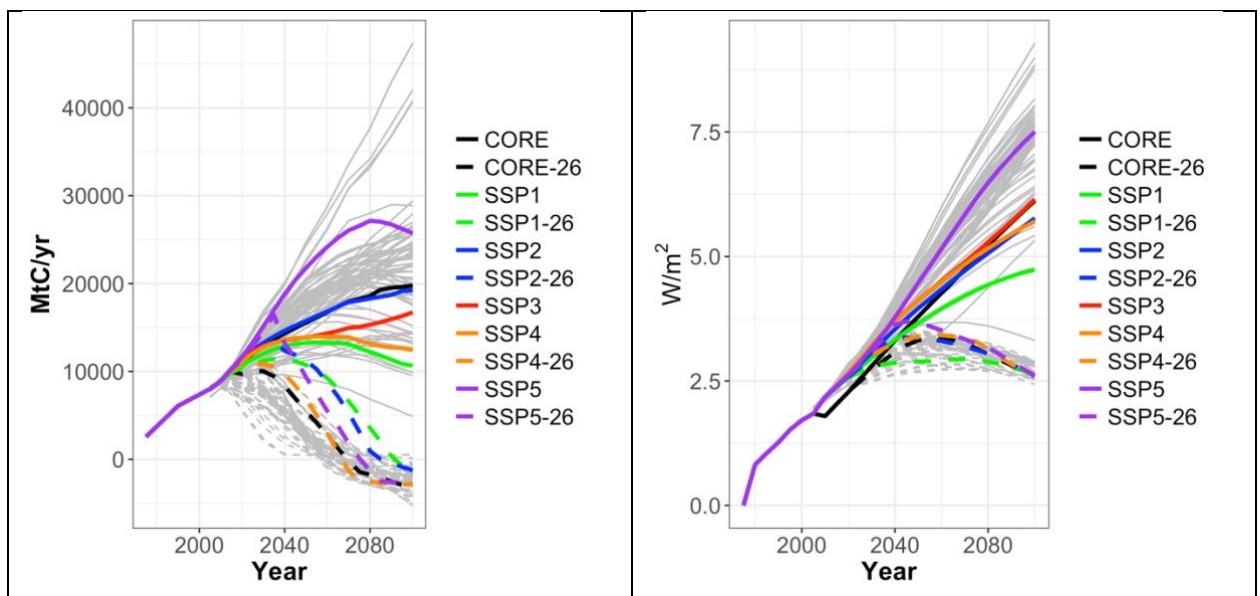

**Figure 16: Global primary energy (top left), global cropland (top right), global fossil fuel and industrial CO₂ emissions (bottom left), and total radiative forcing (bottom right).** Solid gray lines are no climate policy scenarios from the AR5 database (Clarke et al., 2014), i.e., scenarios with a carbon price of $0 in 2100. Dashed gray lines are scenarios in the AR5 database that are roughly consistent with 2.6 W/m²; specifically, we include any scenario with 2100 radiative forcing below 2.9 W/m². Black and colored lines are the GCAM v5.1 scenarios, with color indicating the socioeconomic scenario and line type indicating the climate policy (no policy = solid; 2.6 W/m² = dashed).

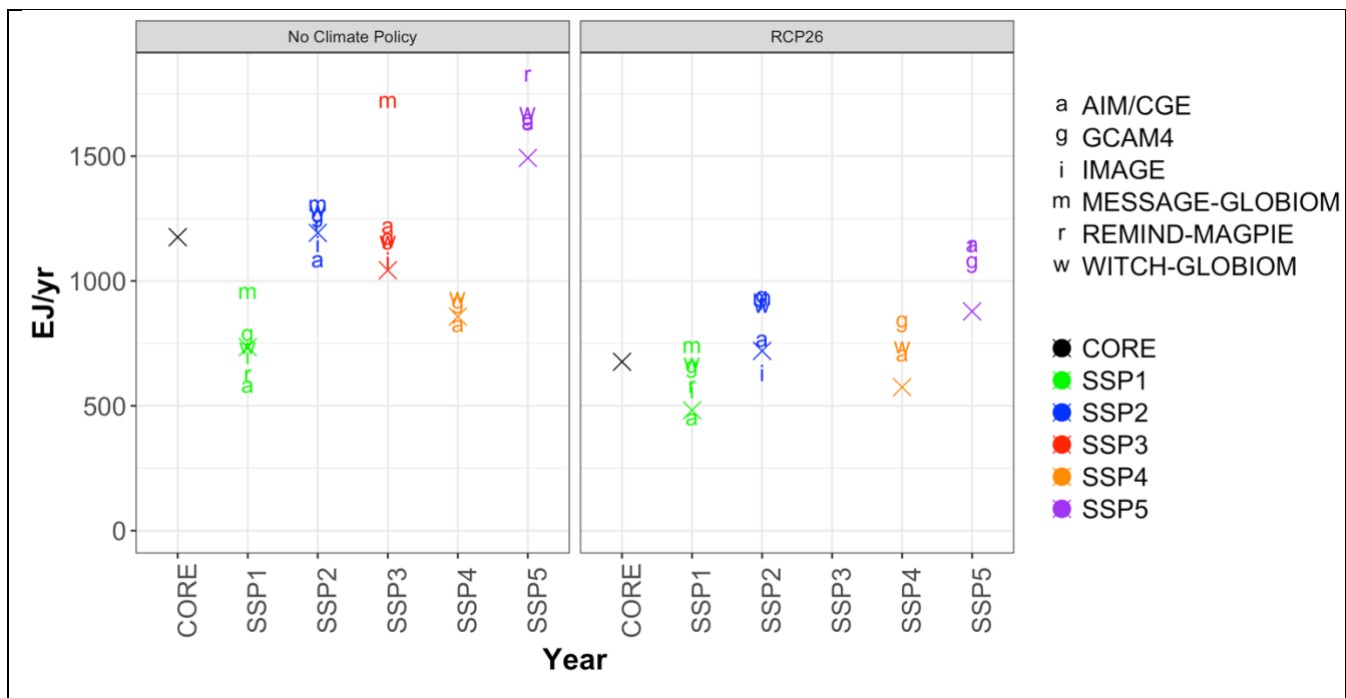

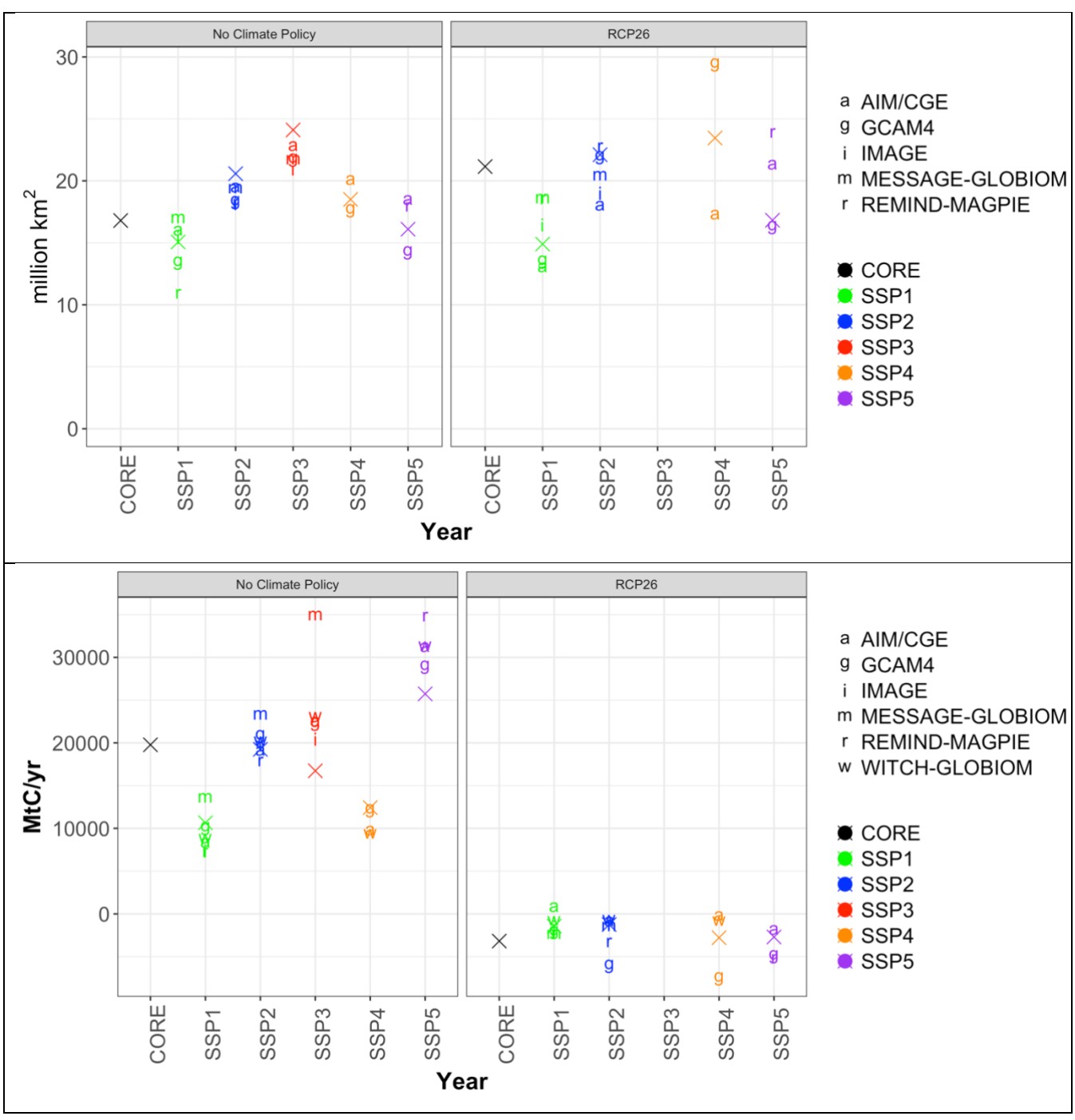

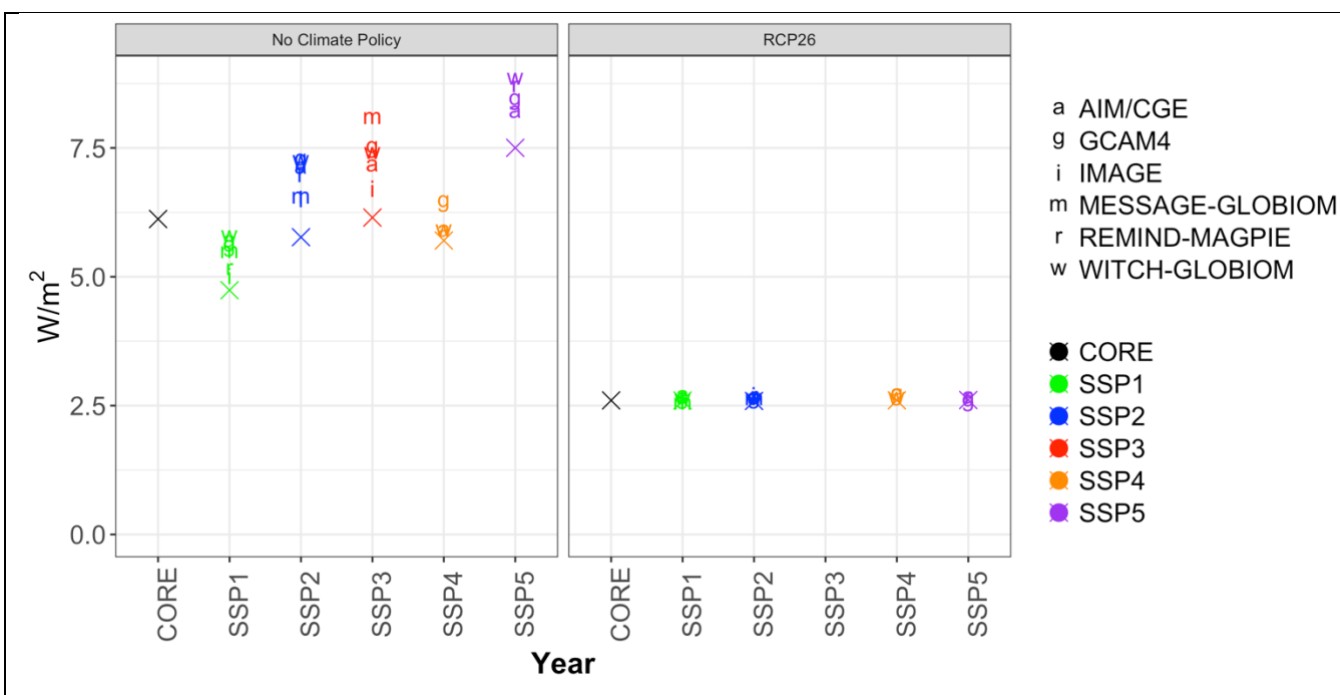

**Figure 17: Global primary energy (top), global cropland (2nd from top), global fossil fuel and industrial CO₂ emissions (2nd from bottom), and total radiative forcing (bottom) in 2100.** GCAM v5.1 results are shown with "X". Data from the SSP database (Riahi et al., 2017a) are shown as lower case letters. Colors indicate SSP. Left panel is the no climate policy results; right panel show scenarios that limit 2100 radiative forcing to 2.6 W/m².