# Peer review of "GCAM v5.1: Representing the linkages between energy, water, land, climate, and economic systems"

_Geoscientific Model Development, 2018_

## Referee Comment (RC1) · Anonymous Referee #1 · 29 Oct 2018

General comments:

The authors provide a comprehensive, clear, and succinct description and documentation of the GCAM v5.1 model. They include eleven scenarios that span a wide range of socioeconomic and climate policy assumptions to illustrate the results. The manuscript provides a valuable resource for potential future users of the model, as well as other researchers wanting to understand the modeling structure, inputs, outputs, and data sources for comparison to other modeling frameworks.

Scientific significance: While this article does not describe and document an entirely new modeling framework, the GCAM model and related analyses make a major con-

tribution to our understanding of the linkages between energy, land, water, climate and socioeconomic systems. It is a widely used and widely cited modeling framework. Version 5.1 appears to represent a substantial advance and update in the data sources, individual model components, as well as the linkages between the key systems. I do have some suggestions, however, to improve the discussion of what is new relative to earlier versions.

Scientific quality: The approach and results are robust. Data sources are valid and well documented. The modeling framework is compared to other models such as IMAGE and MESSAGE (Section 1), and the results of scenarios are compared with other projections (Section 4.3.7) and to historical data (Section 4.7.1.)

Scientific reproducibility: The GCAM model is an open source model with all code and inputs available on GitHub, as well as users guides and trainings. Each individual model component is concisely documented and described, with resources for additional information as needed. The authors provide information on how to obtain the model in the manuscript.

Presentation quality: The authors do an excellent job succinctly and clearly describing the model structure and data sources, and summarizing the key results of their SSP and 2.6 W/m2 scenarios.

Specific comments:

page 2, lines 14-15: Can you be more specific or quantify what is meant by "computationally inexpensive." Very briefly, what are the general system requirements and associated run time for a scenario (e.g., run on the order of minutes, hours?) Can you give the reader a ballpark idea of the run time, say with a standard desktop/laptop versus a higher-end computing cluster? You highlight the growing complexity and computational requirements at the end of the paper, but what does it require now for this model version?

page 3: In this section, I would like to see some additional discussion of what the critical updates are to version 5.1 relative to earlier versions. The authors briefly touch on the differences in the SSP results due to updates in technology costs relative to work published in Muratori (2017). However, can you highlight here, and then briefly summarize in the discussion, the most significant updates to this version of GCAM? I would suggest including that early on in Section 2, or flagging major changes from previous GCAM versions in the individual sections (energy, land, etc) where appropriate. The documentation is very comprehensive, but it was often unclear what was new versus what has been in the model database in previous versions. Please be more explicit about what is new.

page 4, lines 8-17: I would like to see a bit more detailed discussion and additional specifics on the share weights in terms of their role in both model calibration, but also to what extent these can constrain future technology and market shifts. This could be here, or in the individual sections. This may be something more for the discussion section at the end, but given the long time horizon of the model, to what extent are the share weights relaxed or overwritten for future periods. The authors note that these are "on occasion over-written." But, how would user know where and when share weight may be constraining results versus where they can be overwritten to enable more transformative market shifts? How does a user know when a share constraint may be constraining the model in way that perhaps make it harder to reach deep decarbonization targets, for example? These are important model levers and are often not documented with the same rigor and transparency as something like technology cost, for example.

page 6, lines 32-34: I would suggest flagging that this approach, while a fine approximation at a general level, does not capture the changes in emissions factors of non-CO2 emissions that may be induced by policies and air emissions control technologies that individual regions, countries, etc. may adopt in specific time periods.

page 7, line 6: The resources supply curves are now 20 years old. Are there plans to update these?

page 7, line 8: All other biomass energy is supplied from the land component, correct?

page 7, line 24: Do liquid refining plants include both petroleum and biofuels? If so, are biofuels plants modeled separately?

page 8, line 9: It would be useful to know how many crops are modeled, and maybe list the major commodities.

page 8, line 21-22: Is there a mechanism to represent cropland that is idle/fallow in a particular period?

page 9, line 11: What it meant by bioenergy constraints? Lower or upper volume mandates to meet a renewable or low carbon fuel standard? Or are these done as shares of liquid fuel markets?

page 10, line 3: GCAM models water supply as an unlimited resource, but does the model provide the capability to constrain or price water use. It seems there are prices for municipal water use. Are there any prices, or the ability to price or limit quantities of water for cooling technologies or irrigation?

page 10, lines 11-20: The authors introduce the term blue water. It might be helpful to briefly define the terms blue, as well as green and grey water use. Otherwise, I'd suggest omitting.

page 10, lines 27: There is mention of regions that primarily use seawater for electric power cooling system demands. It is unclear, however, whether the water component of the model differentiates between water sources in terms of groundwater, surface water, or water types, fresh, brackish, saline, reuse. Please clarify if these differences are captured or not. Otherwise, I think it's important to highlight in model results that water withdrawals and consumption include all freshwater as well as seawater, since other energy-water analysis may exclude saline/seawater withdrawals.

page 12, line 25: Nice. The example of bioenergy as an example of a coupled system is well described and highlights ones of the key strengths of this type of model, which

is the interactions between systems.

page 13, line 12: Maybe at the end of this section perhaps briefly highlight other examples where the components are tightly coupled.

page 16, line 5: Please clarify what causes the increases in pollution controls, is this solely due to the changes in emissions factors in response to growth in per capita-GDP?

page 17, line 24-25: The data is from CDIAC, but there is no reference. Please provide a citation or link to the website (this data is being transitioned to a new data archive site, correct?)

page 19, line 10-29: See my earlier comments about summarizing the most significant advances in the version 5.1 of GCAM. I think you could tighten up the discussion in paragraphs 2-4 of the discussion to create some space if you are word limited, and still get those key points across to the reader.

page 27, Figure 2: In the caption, maybe describe what the regions are based upon, e.g., are they based on AEZs? Other than getting across a general sense of the spatial scale this graph doesn't convey too much. Perhaps add some descriptions in the Figure caption regarding the basis for each of the regional breakouts.

page 30, Figure 6: In the caption, do you mean that data are aggregated up for both land allocation and crop types? Or just for land allocation? Are there more crop types than this? See my earlier comment for page 8.

page 32, Figure 10: The letters labels seem unnecessary, (a oil, b natural gas). It would be cleaner if you left them off the legend.

page 34, Figure 12: Can you make the data points bigger? It's hard to make out the ones on the line. Also, wasted space with the lower right quadrant. I'd add something or drop the CDIAC versus GCAM data and just summarize in words.

page 35, Figure 13: This mapping onto the AR5 data is very useful for comparison.

Technical corrections:

In the references, please provide the links for the data from EIA, EPA, FAO AQUASTAT, IBNET, IEA, IHA, etc...

References need a lot of clean up, e.g., Kriegler, this looks to be a thesis. Please cite as such. Macknick et al is missing information, this is a report.

---

## Referee Comment (RC2) · Anonymous Referee #2 · 1 Dec 2018

This manuscript briefly introduced the GCAM model structure, the core systems (socioeconomic, energy, agricultural and land use, water, climate), and the databases. The authors provided 11 scenarios based on the combinations of different socioeconomic and climate policy assumptions, and illustrated the results. Overall, the manuscript was useful for the readers to understand this model, and the data source and references are valid. I would recommend it to be published after the following issues are clarified.

Generally comments

Since the new version of GCAM v5.1 is introduced in this manuscript, a summary of

updates from the previous version is necessary to be provided clearly, in aspects of model structure, databases, linkages, etc.

The 11 scenarios were analyzed mainly based on the global scale. I would suggest the authors to provide some country-level discussion or analysis. I think that will be helpful for us to understand the discrepancy in changes among regions or countries estimated by the model.

Specific comments

Page 2 Line 20, "There are a number of models in the community with similar overall scope to GCAM, although each has a unique structure and focus." Could the authors provide a summary about what the unique ability of GCAM have? Any advantages and disadvantages about all the models?

Page 4 Line 12, "The exact share a given option receives in GCAM depends on the logit exponent and the share weight", What is the difference between logit exponent and the share weight? I would suggest the author to provide the formula to demonstrate how the logit exponent and share weight influence on the decision making.

Page 6 Line 3, "whereas depletable resources are indicated as cumulative resource quantities (in EJ), which are drawn down in each time period." How to determine the total available amount for depletable resources? Is it fixed or changeable?

Page 6 Line 21, "the additional cost is equal to the emissions price multiplied by the amount of emissions of the specified species released per unit of output." How is the emission price calculated in the model for $CO_2$ and non-$CO_2$? Is it related to the control measures of pollutants or their harmful effects? Will it change over time?

Page 6 Line 31, the emission factor is actually related to the control application rate. When control rate is high, the emission factor is low. How was it considered in the model?

Page 7 Line 15, "Final demand sectors include buildings (residential and commercial),

transportation (passenger and freight), and industrial (fertilizer, cement, and general industry) sectors." Have the detailed transportation types been considered, such as aviation, railway, in-land water, shipping, etc.? Will the detail industrial sectors be included in future?

Page 10 Line 2, "water supply is an unlimited resource" The water supply is very limited in some regions, such as desert. Will it be an issue?

Page 19 Line 17, "Research versions of GCAM already include new dynamics such as the effects of climate on water supplies, energy demands, and crop yields." Any references to provide here will be good

Figure 4 and Figure 9, since there are no differences between CORE and CORE-26 (barely see the dashed line), I would suggest just use one series in the plot, for both CORE and CORE-26

Figure 5, two "geothermal" here, i and I

Figure 5 and Figure 10, I would suggest to delete the first letter for each series label

---

## Author Comment (AC1) · 3 Jan 2019

Reviewer comment: General comments: The authors provide a comprehensive, clear, and succinct description and documentation of the GCAM v5.1 model. They include eleven scenarios that span a wide range of socioeconomic and climate policy assumptions to illustrate the results. The manuscript provides a valuable resource for potential future users of the model, as well as other researchers wanting to understand the modeling structure, inputs, outputs, and data sources for comparison to other modeling frameworks.

Scientific significance: While this article does not describe and document an entirely new modeling framework, the GCAM model and related analyses make a major contribution to our understanding of the linkages between energy, land, water, climate and socioeconomic systems. It is a widely used and widely cited modeling framework. Version 5.1 appears to represent a substantial advance and update in the data sources, individual model components, as well as the linkages between the key systems. I do have some suggestions, however, to improve the discussion of what is new relative to earlier versions.

Scientific quality: The approach and results are robust. Data sources are valid and well documented. The modeling framework is compared to other models such as IMAGE and MESSAGE (Section 1), and the results of scenarios are compared with other projections (Section 4.3.7) and to historical data (Section 4.7.1.)

Scientific reproducibility: The GCAM model is an open source model with all code and inputs available on GitHub, as well as users guides and trainings. Each individual model component is concisely documented and described, with resources for additional information as needed. The authors provide information on how to obtain the model in the manuscript.

Presentation quality: The authors do an excellent job succinctly and clearly describing the model structure and data sources, and summarizing the key results of their SSP and 2.6 W/m2 scenarios.

Author Response: Thank you for the helpful comments.

Author Changes: We have revised the manuscript in response to your comments and that of the other referee.

Specific comments:

Reviewer comment: page 2, lines 14-15: Can you be more specific or quantify what is meant by "computationally inexpensive." Very briefly, what are the general system

requirements and associated run time for a scenario (e.g., run on the order of minutes, hours?) Can you give the reader a ballpark idea of the run time, say with a standard desktop/laptop versus a higher-end computing cluster? You highlight the growing complexity and computational requirements at the end of the paper, but what does it require now for this model version?

Author Response: A single 100-year simulation using GCAM runs in 10-15 minutes on a laptop. More complex options, e.g., limiting radiative forcing to a particular level, requires numerous sequential 100-year simulations increasing the run time.

Author Changes: We have added a footnote indicating runtime: "A single 100-year simulation using GCAM runs in 10-15 minutes on a laptop. More complex options, e.g., limiting radiative forcing to a particular level, requires numerous sequential 100-year simulations increasing the run time."

Reviewer comment: page 3: In this section, I would like to see some additional discussion of what the critical updates are to version 5.1 relative to earlier versions. The authors briefly touch on the differences in the SSP results due to updates in technology costs relative to work published in Muratori (2017). However, can you highlight here, and then briefly summarize in the discussion, the most significant updates to this version of GCAM? I would suggest including that early on in Section 2, or flagging major changes from previous GCAM versions in the individual sections (energy, land, etc) where appropriate. The documentation is very comprehensive, but it was often unclear what was new versus what has been in the model database in previous versions. Please be more explicit about what is new.

Author Response: We have added a new subsection, now numbered 2.4, that summarizes the critical updates between this version of GCAM and GCAM4.

Author Changes: We have added a new subsection:

"Over time, GCAM has evolved to incorporate new features and more detail, such as

more detailed land use (starting with GCAM v3), increased regional resolution (starting with GCAM v4), and incorporating water demand (starting with v5). The most recent updates (relative to GCAM v4) include: - Incorporating water demands, - Changing the land regions to be based on water basins, instead of agro-ecological zones, - Including multiple agricultural management practices, which enables intensification, - Including five alternative socioeconomic pathways, - Updating to a newer version of the climate model, and - Including a new data processing system."

Reviewer comment: page 4, lines 8-17: I would like to see a bit more detailed discussion and additional specifics on the share weights in terms of their role in both model calibration, but also to what extent these can constrain future technology and market shifts. This could be here, or in the individual sections. This may be something more for the discussion section at the end, but given the long time horizon of the model, to what extent are the share weights relaxed or overwritten for future periods. The authors note that these are "on occasion over-written." But, how would user know where and when share weight may be constraining results versus where they can be over-written to enable more transformative market shifts? How does a user know when a share constraint may be constraining the model in way that perhaps make it harder to reach deep decarbonization targets, for example? These are important model levers and are often not documented with the same rigor and transparency as something like technology cost, for example.

Author Response: The reviewer is correct that share weights do influence future technology deployment. One important note regarding share weights is that they are a variable that model users can adjust as they see fit to create different scenarios from those released in GCAM 5.1, just as technology costs and performance can be adjusted by users.

For simplicity, our general philosophy for share weights in GCAM 5.1 is to maintain them at their calibrated values, which ensure the model replicates history. We do, however, make adjustments to this approach in at least two circumstances: 1. We

consider that the past is not a good analog for the future, as with emerging technologies (e.g., solar, wind) where information barriers or lack of infrastructure may prevent their adoption today, but those factors will likely be ameliorated with time, or 2. The specific scenario being produced necessitates changes to the share weights. For example, we have adjusted the renewable share weights in the SSP1 scenario to reflect stronger preferences for wind and solar.

It is also important to note that one goal in GCAM 5.1 development is transparency regarding the many assumptions in the model. We have constructed an approach to data and assumption development that allows users and those interpreting the model to easily find key assumptions, such as those regarding share weights, in the input files or in the GCAM data system.

Author Changes: We have added a few sentences to the manuscript elaborating on how share weights are chosen/used: "The general philosophy in GCAM is to maintain share weights at their calibrated values (which ensure the model replicates history) unless: 1. We consider that the past is not a good analog for the future, as with emerging technologies (e.g., solar, wind) where information barriers or lack of infrastructure may prevent their adoption today, but those factors will likely be ameliorated with time, or 2. The specific scenario being produced necessitates changes to the share weights (e.g., solar and wind in the SSP1)."

Reviewer comment: page 6, lines 32-34: I would suggest flagging that this approach, while a fine approximation at a general level, does not capture the changes in emissions factors of non-CO2 emissions that may be induced by policies and air emissions control technologies that individual regions, countries, etc. may adopt in specific time periods.

Author Response: Yes, while this approach doesn't explicitly represent individual policies, it is designed to capture the general trend that emissions policies and emissions control technologies increase with income.

Author Changes: We have added a sentence with this caveat: "This approach is designed to capture general trends in emissions factors, but does not explicitly represent individual technologies or policies that may be adopted."

Reviewer comment: page 7, line 6: The resources supply curves are now 20 years old. Are there plans to update these?

Author Response: We model resources, not reserves, and estimates of resources tend to change less over time. We have looked at some updated curves (Rogner et al. 2012) and there is little change in oil & gas resources. Coal supply curves in this publication seemed to only include reserves, a much smaller number than resources, making it difficult to update consistently.

Author Changes: No changes made.

Reviewer comment: page 7, line 8: All other biomass energy is supplied from the land component, correct?

Author Response: Yes, and these sources of biomass are discussed later in this section.

Author Changes: We have added a sentence clarifying this: "other sources of biomass energy are supplied by the land component and discussed later."

Reviewer comment: page 7, line 24: Do liquid refining plants include both petroleum and biofuels? If so, are biofuels plants modeled separately?

Author Response: Yes, we include several different types of refineries, including petroleum, biofuels, coal-to-liquids, and gas-to-liquids. Biofuels plants are modeled separately, with separate cost and performance characteristics.

Author Changes: We have added additional information on refineries, and energy transformation in general: "For example, the energy transformation sectors include a variety of technologies representing different electricity generation facilities (including

different fuel sources and technologies), different refineries (e.g., petroleum, bioliquids, coal-to-liquids, gas-to-liquids), different gas processing facilities, and different hydrogen production facilities. Each technology is specified with a different set of inputs, costs, and performance characteristics."

Reviewer comment: page 8, line 9: It would be useful to know how many crops are modeled, and maybe list the major commodities.

Author Response: We include all crops represented in the FAO database, but aggregate them to 15 commodity categories.

Author Changes: We have added a sentence at the beginning of the land section with this information: "GCAM includes all commodities reported by the FAO, but aggregates them into 15 commodity classes (e.g., Corn, Rice, Wheat, SugarCrop, OilCrop, Forest, Pasture, etc.)."

Reviewer comment: page 8, line 21-22: Is there a mechanism to represent cropland that is idle/fallow in a particular period?

Author Response: We do include "other arable land" which includes fallow cropland. However, we do not explicitly include rotations that alternate fallow with crops.

Author Changes: We have added "other arable land" to the list of land cover types.

Reviewer comment: page 9, line 11: What it meant by bioenergy constraints? Lower or upper volume mandates to meet a renewable or low carbon fuel standard? Or are these done as shares of liquid fuel markets?

Author Response: GCAM can do this in any of the ways mentioned. We can impose a lower or upper bound, specified in EJ, for bioenergy in any region or time period. We can also impose a policy that specifies a share of liquid fuels produced by bioenergy or sets an upper bound on the amount of bioliquids consumed.

Author Changes: We have added a parenthetical note listing these options: "(e.g.,

lower or upper bounds on total bioenergy consumption or the share of bioenergy in liquid fuels)"

Reviewer comment: page 10, line 3: GCAM models water supply as an unlimited resource, but does the model provide the capability to constrain or price water use. It seems there are prices for municipal water use. Are there any prices, or the ability to price or limit quantities of water for cooling technologies or irrigation?

Author Response: Yes, we include a price of water that can be changed easily in the model. Additionally, the capability exists to include a constraint on water quantity. We intend to release files that facilitate these sorts of experiments in the next release.

Author Changes: We have added a sentence to this paragraph on prices: "The price for this resource can be specified by the user."

Reviewer comment: page 10, lines 11-20: The authors introduce the term blue water. It might be helpful to briefly define the terms blue, as well as green and grey water use. Otherwise, I'd suggest omitting.

Author Response: We have opted to omit the term from the paper.

Author Changes: We have removed the references to blue and green water from this paragraph.

Reviewer comment: page 10, lines 27: There is mention of regions that primarily use seawater for electric power cooling system demands. It is unclear, however, whether the water component of the model differentiates between water sources in terms of groundwater, surface water, or water types, fresh, brackish, saline, reuse. Please clarify if these differences are captured or not. Otherwise, I think it's important to highlight in model results that water withdrawals and consumption include all freshwater as well as seawater, since other energy-water analysis may exclude saline/seawater withdrawals.

Author Response: In the version described in this paper, GCAM does not distinguish between types of water, lumping all into a single "resource". It is possible to separate

water supply into freshwater, groundwater, seawater, etc. We are working on facilitating this separation and will release this capability to the community at a later date.

Author Changes: We have modified the sentence describing water resources in response to this comment: "In GCAM v5.1, water supply is an unlimited resource, including all sources of water (e.g., freshwater, groundwater, seawater)."

Reviewer comment: page 12, line 25: Nice. The example of bioenergy as an example of a coupled system is well described and highlights ones of the key strengths of this type of model, which is the interactions between systems.

Author Response: Thank you!

Author Changes: No changes made

Reviewer comment: page 13, line 12: Maybe at the end of this section perhaps briefly highlight other examples where the components are tightly coupled.

Author Response: We have added some more examples.

Author Changes: We have added an additional paragraph: "Fertilizer is another example of a tightly coupled system, with its production determined by the energy system and consumption determined by the land system. Additionally, many other aspects of GCAM create direct or indirect linkages among sectors (e.g., water demand is linked to the energy and agricultural production, climate is linked to emissions produced by the energy and land systems)."

Reviewer comment: page 16, line 5: Please clarify what causes the increases in pollution controls, is this solely due to the changes in emissions factors in response to growth in per capita-GDP?

Author Response: Yes, this due to the decline in emissions factors as a result of increased GDP.

Author Changes: We have modified this sentence to clarify: "increases in pollution

controls induced by rising incomes."

Reviewer comment: page 17, line 24-25: The data is from CDIAC, but there is no reference. Please provide a citation or link to the website (this data is being transitioned to a new data archive site, correct?)

Author Response: Yes, the data is moving to a new site.

Author Changes: We have added a link to the website where the data is currently.

Reviewer comment: page 19, line 10-29: See my earlier comments about summarizing the most significant advances in the version 5.1 of GCAM. I think you could tighten up the discussion in paragraphs 2-4 of the discussion to create some space if you are word limited, and still get those key points across to the reader.

Author Response: See above

Author Changes: We have added an additional subsection in section 2, as described above. Additionally, we have added a sentence to this section: "GCAM v5.1 describes a new version of GCAM, including several major enhancements from previous versions (e.g., water demand, multiple agricultural management practices, new land regions, new data system, newer climate model, alternative socioeconomic pathways)."

Reviewer comment: page 27, Figure 2: In the caption, maybe describe what the regions are based upon, e.g., are they based on AEZs? Other than getting across a general sense of the spatial scale this graph doesn't convey too much. Perhaps add some descriptions in the Figure caption regarding the basis for each of the regional breakouts.

Author Response: We have added information on the regions to the figure caption.

Author Changes: We have added this to the caption: "Regions are based on geopolitical boundaries for energy and economy, on water basins for water, and on a combination of geopolitical boundaries and water basins for land."

Reviewer comment: page 30, Figure 6: In the caption, do you mean that data are aggregated up for both land allocation and crop types? Or just for land allocation? Are there more crop types than this? See my earlier comment for page 8.

Author Response: This note only applies to land allocation. The only crops missing from the top row are biomass and Forest, which are modeled in a different unit and thus cannot be easily included here. (Note: we had erroneously included forest in this figure in the initial submission, but have removed it now)

Author Changes: We have clarified the aggregation and the crops in the caption: "Note bioenergy and forest are excluded from agricultural production (top) as they are modeled in different units, EJ/yr and m3/yr, respectively. Land cover data (bottom) are aggregated..."

Reviewer comment: page 32, Figure 10: The letters labels seem unnecessary, (a oil, b natural gas). It would be cleaner if you left them off the legend.

Author Response: Thank you for the suggestion.

Author Changes: We have removed these letters.

Reviewer comment: page 34, Figure 12: Can you make the data points bigger? It's hard to make out the ones on the line. Also, wasted space with the lower right quadrant. I'd add something or drop the CDIAC versus GCAM data and just summarize in words.

Author Response: Thank you for the suggestion.

Author Changes: We have improved the figure in response to this comment.

Reviewer comment: page 35, Figure 13: This mapping onto the AR5 data is very useful for comparison.

Author Response: Thank you!

Author Changes: No changes made.

[Figure]

Technical corrections: Reviewer comment: In the references, please provide the links for the data from EIA, EPA, FAO AQUASTAT, IBNET, IEA, IHA, etc...

Author Response: We have added these links.

Author Changes: We have added links within each citation in the reference section.

Reviewer comment: References need a lot of clean up, e.g., Kriegler, this looks to be a thesis. Please cite as such. Macknick et al is missing information, this is a report.

Author Response: We have cleaned up the references, including more information on URLs, DOI, Universities, etc. where appropriate.

Author Changes: We have added additional information to the Kriegler citation (university, department, URL) and the Macknick citation (DOI, city).

Anonymous Referee #2

Reviewer comment: This manuscript briefly introduced the GCAM model structure, the core systems (socioeconomic, energy, agricultural and land use, water, climate), and the databases.

The authors provided 11 scenarios based on the combinations of different socioeconomic and climate policy assumptions, and illustrated the results. Overall, the manuscript was useful for the readers to understand this model, and the data source and references are valid. I would recommend it to be published after the following issues are clarified.

Author Response: Thank you for the helpful comments.

Author Changes: We have revised the paper in response to your comments and those of another referee.

Generally comments Reviewer comment: Since the new version of GCAM v5.1 is introduced in this manuscript, a summary of updates from the previous version is necessary

to be provided clearly, in aspects of model structure, databases, linkages, etc.

Author Response: Thank you for the suggestion.

Author Changes: We have added a new subsection to section 2 based on this comment and that of another reviewer. Additionally, we've added a sentence to the discussion summarizing these updates.

Reviewer comment: The 11 scenarios were analyzed mainly based on the global scale. I would suggest the authors to provide some country-level discussion or analysis. I think that will be helpful for us to understand the discrepancy in changes among regions or countries estimated by the model.

Author Response: Thank you for the helpful suggestion.

Author Changes: We have added three new figures (one each for energy, land, and water results) and paragraphs describing the results in Section 4.

The new energy text: "Energy consumption varies across region, in terms of both total consumption and fuel mix (Figure 6). Furthermore, these regional differences change over time due to differences in socioeconomic growth across regions, so the largest consumers today are not the largest consumers in the future. For example, the USA and China have the highest primary energy consumption in 2010, with 86 and 102 EJ/yr, respectively. In 2100, however, India and Africa_Western have the highest in energy consumption in both the CORE (164 and 127 EJ/yr, respectively) and CORE-26 scenarios (75 and 86 EJ/yr, respectively). For fuel mix, there are regional differences in the share of fossil fuels used in 2010, with much lower shares in Africa_Western and Africa_Eastern than the rest of the world. However, in 2100, the biggest differences are across scenarios and not across regions, with fossil fuel consumption ranging from 70-95% of total primary energy in the CORE scenario and much lower use in the CORE-26."

The new land text: "There are significant differences in land use across regions (Figure

8). However, regions that have large shares of cropland today (e.g., India, Europe, China, USA MidWest, Argentina) also have large shares of cropland in the future in both the CORE and CORE-26 scenarios. In the CORE scenario, bioenergy land is spread throughout the world's agricultural producing regions with only 16 of the 384 regions in GCAM devoting more than 10% of their land to bioenergy and only 1 very small region in Southeast Asia devoting more than 20%. In the CORE-26, higher amounts of bioenergy land are required, resulting in shares of bioenergy land ranging from 0% to 58%. Note that some of the regions with large shares of bioenergy land are small in size. The largest amounts of bioenergy land in absolute value are in the Nile River basin in Africa_Eastern and the Niger River basin in Africa_Western, with 470 and 459 thous km2 of bioenergy land in 2100 in the CORE-26, respectively. Only 10 region/basin combinations have more than 150 thous km2 of bioenergy; these region/basins are found in Africa_Eastern, Africa_Western, India, Canada, and Russia."

The new water text: "Water withdrawals differ significantly across region (Figure 10). The basins with the largest irrigation water withdrawals in 2010 are the Ganges, the Indus, and the Sabarmati. In 2100, the largest irrigation water withdrawals come from these three basins plus the Nile River basin (in both the CORE and CORE-26) and the Arabian Peninsula (in the CORE-26 only). The two largest regions in terms of non-irrigation water withdrawals are the USA and China in 2010 and India and China in 2100 in both the CORE and CORE-26 scenarios."

Specific comments Reviewer comment: Page 2 Line 20, "There are a number of models in the community with similar overall scope to GCAM, although each has a unique structure and focus." Could the authors provide a summary about what the unique ability of GCAM have? Any advantages and disadvantages about all the models?

Author Response: There have been a number of papers that have compared GCAM to other models, both in terms of results and structure. For example, Bauer et al. (2018) provides a comparison between GCAM and 10 other models in terms of model type, regional resolution, technology availability, and basic economic structure. Popp

et al. (2017) provides a comparison of the land component of five models. Rao et al. (2017) provides a comparison of the non-CO2 emissions representation in five models, including information on input data and drivers of emissions.

Author Changes: We have added a sentence referring the reader to these comparisons: "Several recent papers provide comparisons of GCAM to these models and many others, including discussions of model structure, input data, and results (Popp et al. 2017; Bauer et al. 2018; Rao et al. 2017)."

Reviewer comment: Page 4 Line 12, "The exact share a given option receives in GCAM depends on the logit exponent and the share weight", What is the difference between logit exponent and the share weight? I would suggest the author to provide the formula to demonstrate how the logit exponent and share weight influence on the decision making.

Author Response: The logit exponent dictates the width of the distribution around cost or profit, influencing the extent to which economics influence shares. The share weight captures unmeasured or uneconomic factors influencing decisions. Larger logit exponents result in decisions more closely tied to the economics, meaning the least cost or highest profit option will take most of the share. With lower logit exponents, the share weights have a stronger influence; for example, with a logit exponent of zero shares will be determined solely by the share weights.

Author Changes: We have added the share formula to the text to help clarify this point.

Reviewer comment: Page 6 Line 3, "whereas depletable resources are indicated as cumulative resource quantities (in EJ), which are drawn down in each time period." How to determine the total available amount for depletable resources? Is it fixed or changeable?

Author Response: The total amount of a depletable resource available for the entire 100-year simulation is fixed. In each period, the amount remaining is updated to reflect

any consumption in the prior time period.

Author Changes: We have revised this paragraph to be clearer: "Resources may be renewable (e.g., wind, solar), or depletable (e.g., fossil fuels and uranium). Renewable resource supply curves are indicated in EJ per year, whereas depletable resources are indicated as cumulative resource quantities (in EJ), which are drawn down in each time period as each resource is consumed. Resource costs, including depletion-related increases in fossil resource prices, may be counter-acted by exogenous technical change, which lowers extraction costs."

Reviewer comment: Page 6 Line 21, "the additional cost is equal to the emissions price multiplied by the amount of emissions of the specified species released per unit of output." How is the emission price calculated in the model for CO2 and non-CO2? Is it related to the control measures of pollutants or their harmful effects? Will it change over time?

Author Response: The emissions price is determined by the type of policy imposed and can change over time. In a reference or no policy scenario, the emissions price is zero for all gases. In the 2.6 scenarios shown in this paper, there is a price on CO2 chosen to ensure radiative forcing is limited to 2.6 W/m2 in 2100.

Author Changes: We have added an additional clarifying sentence: "Emissions prices can be exogenously specified or generated by the model if a constraint or target is imposed; these prices can vary across time, region, and gas."

Reviewer comment: Page 6 Line 31, the emission factor is actually related to the control application rate. When control rate is high, the emission factor is low. How was it considered in the model?

Author Response: Both are possible. The user can read in emissions factors directly that implicitly reflect control rates. Or, a control rate can be calculated by the model and used to modify the emissions factor. The default approach is the latter, where the

control rate is linked to GDP per capita for pollutants and CO2 prices for GHGs.

Author Changes: We have added a clarification on control rates to this section: "In future years the emission factor may evolve as control rates change in response to growth in per-capita GDP and/or carbon pricing."

Reviewer comment: Page 7 Line 15, "Final demand sectors include buildings (residential and commercial), transportation (passenger and freight), and industrial (fertilizer, cement, and general industry) sectors." Have the detailed transportation types been considered, such as aviation, railway, in-land water, shipping, etc.? Will the detail industrial sectors be included in future?

Author Response: Yes, we have included detailed transportation types, including aviation, rail, in-land water, and shipping. For industrial, we have currently separated cement and fertilizer production. We are continually enhancing GCAM, but if/when more detailed industrial sectors will be included is uncertain and depends on a range of priorities.

Author Changes: We have added a parenthetical note clarifying the detail on transportation: "(passenger and freight, including road, rail, air, and shipping)"

Reviewer comment: Page 10 Line 2, "water supply is an unlimited resource" The water supply is very limited in some regions, such as desert. Will it be an issue?

Author Response: There are studies that show that water scarcity could be a problem in many regions, see for example Hejazi et al. (2014). We are working on developing constraints on water supply that will be included in future model releases.

Author Changes: We have added a citation to the Hejazi et al. article in response to this comment and the next comment.

Reviewer comment: Page 19 Line 17, "Research versions of GCAM already include new dynamics such as the effects of climate on water supplies, energy demands, and crop yields." Any references to provide here will be good

Author Response: We have added references.

Author Changes: We have added references.

Reviewer comment: Figure 4 and Figure 9, since there are no differences between CORE and CORE-26 (barely see the dashed line), I would suggest just use one series in the plot, for both CORE and CORE-26

Author Response: Thank you for the suggestion.

Author Changes: We have removed the additional series and renamed the current series "CORE and CORE-26".

Reviewer comment: Figure 5, two "geothermal" here, i and I

Author Response: Thank you for catching this issue.

Author Changes: We have removed the letters, so there is only a single "geothermal" now.

Reviewer comment: Figure 5 and Figure 10, I would suggest to delete the first letter for each series label

Author Response: We have removed the letters.

Author Changes: We have removed the letters.

References Bauer, N., and Coauthors, 2018: Global energy sector emission reductions and bioenergy use: overview of the bioenergy demand phase of the EMF-33 model comparison. Clim. Change, doi:10.1007/s10584-018-2226-y. http://link.springer.com/10.1007/s10584-018-2226-y.

Hejazi, M. I., and Coauthors, 2014: Integrated assessment of global water scarcity over the 21st century under multiple climate change mitigation policies. Hydrol. Earth Syst. Sci., 18, 2859–2883, doi:10.5194/hess-18-2859-2014.

Popp, A., and Coauthors, 2017: Land-use futures in the shared socio-economic pathways. Glob. Environ. Chang., 42, doi:10.1016/j.gloenvcha.2016.10.002.

Rao, S., and Coauthors, 2017: Future air pollution in the Shared Socio-economic Pathways. Glob. Environ. Chang., 42, 346–358, doi:https://doi.org/10.1016/j.gloenvcha.2016.05.012. http://www.sciencedirect.com/science/article/pii/S0959378016300723.

Rogner, H.-H., and Coauthors, 2012: Chapter 7 - Energy Resources and Potentials. Global Energy Assessment - Toward a Sustainable Future, Cambridge University Press, Cambridge, UK and New York, NY, USA and the International Institute for Applied Systems Analysis, Laxenburg, Austria, 423–512 http://www.globalenergyassessment.org.